# Probabilistic seasonal dengue forecasting in Vietnam: A modelling study using superensembles

Felipe J. Colón-González[1,2,3,4]*, Leonardo Soares Bastos[1,3,5], Barbara Hofmann[6], Alison Hopkin[6], Quillon Harpham[6], Tom Crocker[7], Rosanna Amato[7], Iacopo Ferrario[6], Francesca Moschini[6], Samuel James[6], Sajni Malde[6], Eleanor Ainscoe[6], Vu Sinh Nam[8], Dang Quang Tan[9], Nguyen Duc Khoa[9], Mark Harrison[7], Gina Tsarouchi[6], Darren Lumbroso[6], Oliver J. Brady[1,2☯], Rachel Lowe[1,2,3☯]

1 Centre for the Mathematical Modelling of Infectious Diseases, London School of Hygiene & Tropical Medicine, London, United Kingdom, 2 Department of Infectious Disease Epidemiology, Faculty of Epidemiology and Population Health, London School of Hygiene & Tropical Medicine, London, United Kingdom, 3 Centre on Climate Change and Planetary Health, London School of Hygiene & Tropical Medicine, London, United Kingdom, 4 Tyndall Centre for Climate Change Research, University of East Anglia, Norwich, United Kingdom, 5 Scientific Computing Programme, Oswaldo Cruz Foundation (Fiocruz), Rio de Janeiro, 6 HR Wallingford, Wallingford, Oxfordshire, United Kingdom, 7 Met Office, Exeter, Devon, United Kingdom, 8 National Institute of Hygiene and Epidemiology, Hanoi, Vietnam, 9 General Department of Preventive Medicine, Hanoi, Vietnam

☯ These authors contributed equally to this work.
* Felipe.Colon@lshtm.ac.uk

**Data Availability Statement:** The data underlying the results presented in the study are available from: Dengue data: Dr Pham Hung, Head of Disease Control Department, Vietnam. Email:

## Abstract

### Background

With enough advanced notice, dengue outbreaks can be mitigated. As a climate-sensitive disease, environmental conditions and past patterns of dengue can be used to make predictions about future outbreak risk. These predictions improve public health planning and decision-making to ultimately reduce the burden of disease. Past approaches to dengue forecasting have used seasonal climate forecasts, but the predictive ability of a system using different lead times in a year-round prediction system has been seldom explored. Moreover, the transition from theoretical to operational systems integrated with disease control activities is rare.

### Methods and findings

We introduce an operational seasonal dengue forecasting system for Vietnam where Earth observations, seasonal climate forecasts, and lagged dengue cases are used to drive a superensemble of probabilistic dengue models to predict dengue risk up to 6 months ahead. Bayesian spatiotemporal models were fit to 19 years (2002–2020) of dengue data at the province level across Vietnam. A superensemble of these models then makes probabilistic predictions of dengue incidence at various future time points aligned with key Vietnamese decision and planning deadlines. We demonstrate that the superensemble generates more accurate predictions of dengue incidence than the individual models it incorporates across a

hungvncdc@gmail.com Precipitation The precipitation dataset used is a derivative work from NASA GPM and NASA TRRM, https://www.usa.gov/government-works. Wind Speed The wind speed dataset used is a derivative work from NOAA Climate Forecast System Version 2 (CFSv2) Operational Analysis, https://www.usa.gov/government-works. Land Cover The land cover dataset used is a derivative work from Land Cover, © ESA Climate Change Initiative - Land Cover led by UCLouvain (2017). https://www.esa-landcover-cci.org/?q=node/164 Population Population data was used from: WorldPop (www.worldpop.org - School of Geography and Environmental Science, University of Southampton; Department of Geography and Geosciences, University of Louisville; Departement de Geographie, Universite de Namur) and Center for International Earth Science Information Network (CIESIN), Columbia University (2018). Global High Resolution Population Denominators Project - Funded by The Bill and Melinda Gates Foundation (OPP1134076). https://dx.doi.org/10.5258/SOTON/WP00645. Licenced under Creative Commons Attribution 4.0 International License (http://creativecommons.org/licenses/by/4.0). GPW Population Count, v4.10 (2000, 2005, 2010, 2015, 2020). Licenced under Creative Commons Attribution 4.0 International License (http://creativecommons.org/licenses/by/4.0). UN WPP UN WPP-Adjusted Population Count, v4.10 (2000, 2005, 2010, 2015, 2020). Center for International Earth Science Information Network (CIESIN), Columbia University. 2017. Gridded Population of the World, Version 4 (GPWv4): Population Count Adjusted to Match 2015 Revision of UN WPP Country Totals, Revision 10. Palisades, NY: NASA Socioeconomic Data and Applications Center (SEDAC). https://doi.org/10.7927/H4JQ0XZW. Accessed 16/09/2019. Humidity The humidity dataset used is a derivative work from NASA MODIS/Terra Total Precipitable Water Vapor 5-Min L2 Swath 1km and 5km – NRT, https://www.usa.gov/government-works. Temperature The temperature dataset used is a derivative work from NASA MODIS/Terra Land Surface Temperature/Emissivity Daily L3 Global 1km SIN Grid V006 and MODIS/Aqua Land Surface Temperature/Emissivity Daily L3 Global 1km SIN Grid V006, https://www.usa.gov/government-works.

**Funding:** This work was funded by the UK Space Agency (https://www.gov.uk/government/organisations/uk-space-agency) Dengue forecasting MOdel Satellite-based System (D-MOSS) awarded to GT, DL, MH, OJB and RL. FJCG, LSB, BH, QH, AH, TC, RA, IF, FM, SJ, SM, and EA were supported by the D-MOSS grant. OJB

suite of time horizons and transmission settings. Using historical data, the superensemble made slightly more accurate predictions (continuous rank probability score [CRPS] = 66.8, 95% CI 60.6–148.0) than a baseline model which forecasts the same incidence rate every month (CRPS = 79.4, 95% CI 78.5–80.5) at lead times of 1 to 3 months, albeit with larger uncertainty. The outbreak detection capability of the superensemble was considerably larger (69%) than that of the baseline model (54.5%). Predictions were most accurate in southern Vietnam, an area that experiences semi-regular seasonal dengue transmission. The system also demonstrated added value across multiple areas compared to previous practice of not using a forecast. We use the system to make a prospective prediction for dengue incidence in Vietnam for the period May to October 2020. Prospective predictions made with the superensemble were slightly more accurate (CRPS = 110, 95% CI 102–575) than those made with the baseline model (CRPS = 125, 95% CI 120–168) but had larger uncertainty. Finally, we propose a framework for the evaluation of probabilistic predictions. Despite the demonstrated value of our forecasting system, the approach is limited by the consistency of the dengue case data, as well as the lack of publicly available, continuous, and long-term data sets on mosquito control efforts and serotype-specific case data.

## Conclusions

This study shows that by combining detailed Earth observation data, seasonal climate forecasts, and state-of-the-art models, dengue outbreaks can be predicted across a broad range of settings, with enough lead time to meaningfully inform dengue control. While our system omits some important variables not currently available at a subnational scale, the majority of past outbreaks could be predicted up to 3 months ahead. Over the next 2 years, the system will be prospectively evaluated and, if successful, potentially extended to other areas and other climate-sensitive disease systems.

## Author summary

### Why was this study done?

- A climate-driven dengue early warning system would allow public health decision-makers to design, implement, and target timely interventions to the most at-risk places.
- Dengue is sensitive to changes in temperature, rainfall, humidity, and wind speed. Therefore, a dengue prediction model driven by seasonal climate forecasts offers a valuable tool for predicting dengue risk in advance.

### What did the researchers do and find?

- We developed a superensemble of probabilistic models to predict dengue incidence across Vietnam up to 6 months ahead. The predictive ability of the superensemble was assessed using multiple verification metrics and compared to a baseline model at multiple lead times, seasons, and locations.

was supported by a Wellcome Trust Sir Henry Wellcome Fellowship (206471/Z/17/Z). RL was supported by a Royal Society Dorothy Hodgkin Fellowship. The funders had no role in the study design, data collection and analysis, decision to publish, or preparation of the manuscript.

**Competing interests:** The authors have declared that no competing interests exist.

**Abbreviations:** ARIMA, autoregressive integrative moving average; CRPS, continuous rank probability score; CRPSS, continuous rank probability skill score; D-MOSS, Dengue forecasting MOdel Satellite-based System; DTR, diurnal temperature range; ECMWF, European Centre for Medium-Range Weather Forecasts; GLMM, generalised linear mixed model; LASSO, least absolute shrinkage and selection operator; MADN, median absolute deviation about the median; MAE, mean absolute error; NIHE, National Institute of Hygiene and Epidemiology; RMSE, root mean squared error; SEDAC, Socioeconomic Data and Applications Center; TIHE, Institute of Hygiene and Epidemiology Tay Nguyen; TSCV, time series cross-validation; WHO, World Health Organization.

- We found that the model superensemble generated more accurate predictions than the baseline model 1 to 3 months ahead but not 4 to 6 months ahead.

- The model superensemble, however, was better able to predict outbreaks than the baseline model across all lead times.

- The predictive ability of the model varied with geographic location, forecast horizon, and time of the year and performed best in the peak season in areas experiencing a high level of transmission.

## What do these findings mean?

- While outbreaks of infectious diseases are difficult to predict, particularly several months ahead, model superensembles, which combine multiple climatic drivers with dengue surveillance data at the time of forecast issue date, provide a useful decision-support tool.

- Early warning systems driven by seasonal climate forecasts could shift dengue control towards a more preventative approach, guiding bespoke and targeted public health interventions and a more efficient allocation of scarce resources.

## Introduction

Dengue is a mosquito-transmitted viral infection spread by *Aedes* mosquitoes in urban and peri-urban environments in tropical and subtropical countries [1–3]. About half of the global population is at risk of dengue transmission [4,5]. Dengue infection is characterised by flu-like symptoms that may include sudden intermittent high fever, retro-orbital pain, muscle and joint pain, severe headache, and widespread red skin rash. Its treatment typically includes the supportive care of symptoms. There is no specific antiviral treatment for dengue, and efforts to control transmission focus on controlling vector populations [6]. A live, attenuated, tetravalent dengue vaccine has demonstrated efficacy in 2 large-scale Phase III trials [7] and has now been licensed. However, while the vaccine shows high levels of protection against disease in people with previous exposure to dengue viruses, it may increase the risk of severe dengue if given to seronegative individuals [7]. A lack of reliable and scalable tests for seropositivity currently hinders wider rollout of the Dengvaxia vaccine and so most countries still primarily rely on insecticide-based mosquito control interventions to limit dengue virus transmission [8]. The increasing resistance to insecticides highlights the need for targeted and effective interventions [9].

Vietnam is particularly affected by dengue with an estimated burden of about 2 million yearly infections [5,10], although, on average, only 95,000 cases have been reported annually to the Ministry of Health over the period 2002 to 2020. Underreporting may be due to 2 main factors. First, dengue surveillance in Vietnam is mostly passive, relying on clinical cases reported by patients seeking healthcare [11]. Second, up to 80% of the cases may be asymptomatic or minimally symptomatic and will likely not seek healthcare [12]. The economic impact of dengue in Vietnam is estimated to be US$30 to US$95 million per annum [10,13,14]. Dengue in Vietnam is primarily spread by *Aedes aegypti* and, to a lesser extent, by *Aedes albopictus* [15,16].

In Vietnam, dengue is characterised by strong seasonality and substantial interannual and spatial variability (S1 Fig). Dengue exhibits different behaviour in different parts of the country. In the north, where temperatures are lower than in the rest of the country, most provinces have few or no cases. An exception is Hanoi, which has reported, on average, about 8,700 dengue cases per year over the past 10 years. In central and southern provinces where temperatures are warmer, many provinces report thousands of cases annually, albeit with large interannual variation.

Dengue control in the country primarily involves community engagement and mobilisation to reduce breeding sites and outdoor low-volume insecticide spraying in the vicinity of reported dengue cases to kill adult mosquitoes [11]. One limitation of dengue control measures is that they are essentially reactive, meaning they take place after cases have occurred. Thus, vector control, dengue diagnosis, and reporting all suffer delays which vary across time and space [17]. This situation hinders the timely generation and communication of information on when and where transmission occurs, limiting the ability of health professionals to plan and execute control measures.

If accurate predictions are available, public health decision-makers and planners could design, implement, and target interventions to the most at-risk places in a timely fashion. Disease models driven by Earth observations have been valuable for predicting dengue risk ahead of time, supporting decision-making in multiple settings [18–20]. Climate variation is one of the main drivers of dengue ecology. It is known that temperature regulates the development, biting, reproduction rates, and the spatial distribution of *Aedes* mosquitoes [21–23]. The titre and replication of dengue viruses within mosquitoes is temperature dependent [21,24]. For example, rising temperatures increase dengue transmission to an optimum range of 26 to 29˚C [25], and large diurnal temperature ranges (DTRs; above 20˚C) reduce transmission and increase mosquito mortality [26]. Mosquito survival is also affected by humidity. Research indicated that *A. aegypti* eggs survive across a wide range of humidity and temperature combinations. However, *Ae. albopictus* eggs experience high mortality when relative humidity is lower than 95% if temperatures are above 22˚C [27]. Under high humidity conditions (i.e., above 80%), adult mosquitoes survive and remain active by replenishing transient body fluid depletion with plant sugars and host blood [28]. This behaviour is also observed in lower humidity environments, although to a lesser extent [28]. The effect of precipitation on dengue is related to the creation or flush away of mosquito breeding sites [29]. The effects of wind speed have been seldom explored in the literature. However, research suggests that wind speed reduces the biting activity of mosquitoes, reducing dengue risk [30]. Most of the effects of climate on dengue incidence are delayed by 1 to 2 months due to their effect on the life cycle of both mosquitoes and the dengue virus [31]. Thus, the climatic conditions just before the transmission season starts may be indicative of dengue incidence in the following dengue transmission season [32,33].

Multiple studies have highlighted the potential usefulness of seasonal climate-driven epidemiological surveillance for decision-making and planning [32,34–36]. These studies have used subseasonal (i.e., between 2 weeks and 2 months ahead) forecasts to inform disease models and compute predictions of dengue risk. There has been limited progress in using subseasonal to seasonal climate forecasting to compute prospective forecasts on a routine basis. There are several challenges for implementing operational and sustainable subseasonal (henceforth seasonal) early warning systems [37]. Some of these challenges include the lack of multi-decadal health data sets with which to train and validate seasonal climate-driven early warning systems, the common mismatch of scales between climate data outputs and data used for decision-making, and a general lack of consensus as to how to communicate uncertainties to users [36].

Dengue early warning systems driven by Earth observations and seasonal climate forecasts have been proposed using a range of modelling approaches [38], including autoregressive integrative moving average (ARIMA) [39], point forecasts [32,40], spatiotemporal Bayesian hierarchical models [18,19], least absolute shrinkage and selection operator (LASSO) regression [41,42], and machine learning [43]. Often, models are validated using block cross-validation to select the model specification with the lowest out-of-sample predictive error [18,32]. This approach takes advantage of all available data to make repeated out-of-sample model predictions, which increases the robustness of model adequacy and verification statistics. One drawback of this method is that it does not preserve the time ordering of the data. Also, predictions are computed for some time periods using a model trained on data from a later time period [44].

Previous dengue risk prediction studies have relied on outputs from 1 or 2 competing models [18,32]. However, combining forecasts from multiple competing models into a superensemble may result in more accurate predictions than those from any individual model [38,45]. The use of model superensembles for the development of dengue early warning systems has been seldom explored. Moreover, predictions are typically made for a selected year or month [18,32,34] rather than for a series of lead times into the future, or a whole season [19,41]. In some cases, systems are designed exclusively for research purposes in isolation from relevant stakeholders who may become potential users. This situation is partly due to the difficulties related to the integration of these systems into existing public health procedures and occasional lack of technical expertise.

Typically, dengue early warning systems are based on deterministic models [32,39,40] which may underrepresent heterogeneity and stochastic cessation of transmission. However, decision-makers are increasingly interested in understanding the uncertainties related to the models used to develop decision-support tools and in the probabilities that an event of public health concern may or may not take place [34,46,47]. Spatiotemporal probabilistic models have the advantage of being able to quantify the probability that an event (e.g., an outbreak) may occur at specific times and for specific locations. Public health officials may be more inclined to take action if the probability of observing an outbreak exceeds a certain value [18]. Both modellers and decision-makers should pay attention to and agree on the definition of outbreaks thresholds so that model predictions are a useful guide for planning and decision-making.

In several countries, including Vietnam, outbreaks are defined using a so-called endemic channel [48,49], which corresponds to the mean number of cases per month or season over a long-term period [50]. In Vietnam, endemic channels are defined for each province using the last 5 years of dengue surveillance data. Outbreak years are removed from the computation of the endemic threshold. When dengue cases exceed the mean plus 2 standard deviations, an outbreak is declared. One limitation of this approach is that often, outbreak years are removed arbitrarily or quasi-quantitatively to increase the sensitivity of the outbreak threshold [50]. In areas where dengue incidence is typically low (e.g., less than 10 cases per month), the endemic channel may be frequently exceeded, generating statistical alarms of little public health importance [51]. Despite these limitations, endemic channels are widely used for dengue control decision-making in a variety of countries and provide a practical decision point around which forecasts can be targeted [52,53].

Here, we introduce a superensemble of probabilistic spatiotemporal hierarchical dengue models driven by Earth observations and seasonal climate forecasts. The model framework was codesigned with stakeholders from the World Health Organization (WHO), the United Nations Development Programme, the Vietnamese Ministry of Health, the Pasteur Institute Ho Chi Minh City, the Pasteur Institute Nha Trang, the Institute of Hygiene and Epidemiology

Tay Nguyen (TIHE), and the National Institute of Hygiene and Epidemiology (NIHE). The system is designed to generate monthly estimates of dengue risk across Vietnam (331,210 km$^2$) at the province level ($n$ = 63) in near-real time (i.e., current time minus processing time). The superensemble is used in an expanding window time series cross-validation (TSCV) framework [54] to generate probabilistic dengue forecasts, which allow us to calculate the probability of exceeding predefined dengue outbreak thresholds for a forecast horizon (i.e., lead time) of 1 to 6 months.

The superensemble constitutes the dengue fever component of a forecasting system called D-MOSS (i.e., Dengue forecasting MOdel Satellite-based System). The system operates using a suite of Earth observation data sources from satellites and has its first implementation in Vietnam. Vietnam is divided into 63 provinces which are subdivided into 713 districts and has an estimated population of 95.5 million people. The D-MOSS system produces results at the province level and is accompanied by an assessment of its predictive ability that is applied consistently across the whole of Vietnam. The intention is to give an overall evaluation of the performance of our method, rather than an assessment that pertains to the characteristics of any particular province.

## Materials and methods

The analysis plan was described in a UK Space Agency grant proposal for the International Partnership Programme in September 2017. The aims, objectives, and proposed methodology were developed prior to data analysis and are described in S1 Text. Changes to the analytical plan in response to suggestions from reviewers took place between August and October 2020. These changes include (1) calculating the bias and sharpness of each of the competing models included in the superensemble to decompose the overall prediction error; (2) formulating a climate naïve baseline model that captures the spatial dependency and seasonality for each province and estimates the same seasonal average of dengue incidence each month; (3) modifying the model superensemble specification from Bayesian moving averaging to model stacking based on the continuous rank probability score (CRPS); (4) using weights based on the variance of errors rather than maximum likelihood; and (5) modifying the colour palette and the methodology used for communicating and visualising the outputs from the probabilistic superensemble. Our study did not require a separate ethical approval.

### Dengue surveillance data

Monthly dengue cases were obtained from the Vietnamese Ministry of Health. Data were retrieved for the period August 2002 to December 2019 at the province level ($n$ = 63). Data comprised suspected and confirmed dengue cases, although there was no indication as to how many cases fell within each category. The data set did not contain serotype-specific information.

According to the national guidelines on dengue surveillance, suspected cases are defined as people living in or coming from endemic areas or from areas that have had dengue outbreak foci over the previous 14 days with manifestation of sudden high fever lasting for 2 to 7 days and with at least 2 of the following signs: hemorrhagic manifestation (positive tourniquet test, hypodermic petechiae or ecchymosis, and stump hemorrhage or nasal hemorrhage), headache, anorexia, nausea, vomiting, skin flush, rash, muscular, joint and orbital pains, writhe, unconsciousness, pain in liver area or pain when pressing on the liver area, hepatomegaly, thrombocytopenia, or increased hematocrit. Confirmed cases correspond to those confirmed by laboratory tests using MAC-ELISA, PCR, and NS1, or virus isolation. Over the period 2002 to

2019, there have been no changes in the classification of suspected and confirmed dengue cases in Vietnam.

## Historical demographic and land cover data

Total population per province and per year were retrieved from the Socioeconomic Data and Applications Center (SEDAC) Gridded Population of the World project version 4.11 [55] at a 1 $km^2$ resolution, 5 yearly for the period 2000 to 2020. Intervening years were generated using linear interpolation. The percentage of urban, peri-urban, and rural land cover per province and per year for the period 2002 to 2019 was derived from the ESA CCI Land Cover project [56,57], which describes the land surface into 22 classes at a spatial resolution of 0.00277 degrees.

## Historical Earth observation data

Minimum, maximum, and mean air temperature at 2 metres above ground (˚C) were derived from MODIS daily L3 global land surface temperature products [58] with a spatial resolution of 1 $km^2$. Precipitation amount per day (mm day$^{-1}$) was initially retrieved from from the Tropical Rainfall Measurement Mission [59] at a spatial resolution of 25 $km^2$ up to April 2014. After April 2014, precipitation data were obtained from the Global Precipitation Mission [60] at a spatial resolution of 10 $km^2$. Daytime-specific surface humidity (dimensionless) was calculated using the daytime MODIS L2 water vapour near infrared MOD 5 products [61] with a spatial resolution of 1 $km^2$. Average daily wind speed at 10 metres above ground (m s$^{-1}$) was retrieved from the European Centre for Medium-Range Weather Forecasts (ECMWF) ERA-5 reanalysis [57] for the period 2002 to 2011, at a spatial resolution of 31 $km^2$. After 2011, wind speed data were obtained from the NOAA Climate Forecast System [62] at a 20 $km^2$ resolution. Monthly sea surface temperature anomalies for the Niño region 3.4 (5˚S−−5˚N and 170˚−120˚W) were obtained from the NOAA Center for Weather and Climate Prediction Climate Prediction Center [63] for the period 2002 to 2020.

Earth observation data were obtained as a gridded data set in netcdf format. Sociodemographic data were averaged at the province level using the `rasterstats` module in Python 3.6 [64]. When a polygon covered only part of a grid cell, averages were calculated for the approximate fraction of the cell covered by the polygon (rounded 1/100). The administrative boundaries of each province were defined using a shapefile provided by the Vietnamese General Department of Preventive Medicine, Hanoi.

Climate data were aggregated at the province level using population weighted averages for each month. SEDAC population data, although gridded, had identical values at each pixel falling within a given administrative district (i.e., admin 2 level). SEDAC data could not be used for the computation of population weighted averages as each grid cell within an administrative district would carry the same weight. As a compromise, we used WorldPop [65] estimates for which population estimates vary per pixel. Population data from the WorldPop project [65] at a 100 $m^2$ spatial resolution was used to calculate annual gridded weights for each province using the `rasterstats` module in Python 3.6 [64]. At the time of processing, WorldPop data were only available for the years 2009, 2010, 2015, and 2020. Intervening years were calculated using linear interpolation. Province-specific population-weighted averages were calculated as follows:

$$\hat{x} = \frac{\sum_{i=1}^{n} w_i x_i}{\sum_{i=1}^{n} w_i},$$

(1)

where $n$ is the total number of pixels $i$ falling within a given province polygon, $x$ is the value of

each climate predictor at pixel $i$, and $w$ denotes the pixel-specific weights calculated as

$$w_i = \frac{p_i}{N},$$
(2)

where $p_i$ is the pixel-specific population count, and $N$ is the total population count falling into a given province.

There were multiple changes in the administrative boundaries of the provinces over the study period. The last change took place in 2008. Consequently, we froze the administrative boundaries of the country at their 2008 level. Dengue data were allocated to each of the 2008 level administrative boundaries using district-level records. The allocation of dengue cases to each province was conducted by the General Department of Preventive Medicine, Hanoi.

## Seasonal climate forecasts

Seasonal climate forecasts of minimum temperature, maximum temperature, daily precipitation, specific humidity, wind speed, and sea surface temperature anomalies for the Niño region 3.4 were obtained from the UK Met Office Global Seasonal Forecasting System version 5 (GloSea5) [66,67]. GloSea5 comprises 42 ensemble members built around a high-resolution climate prediction model (HadGEM3). Ensemble members differ due to small stochastic physics perturbations provided by the Stochastic Kinetic Energy Backscatter v2 [68]. GloSea5 has a resolution of 0.83 degrees in latitude and 0.56 degrees in longitude for the atmosphere and $0.25 \times 0.25$ degrees for the ocean. GloSea5 has 2 major components, the forecast itself and the associated hindcasts or historical re-forecasts, which are used for calibration and assessment. An evaluation of the predictive ability of the GloSea5 system can be found elsewhere [66]. S2 Fig shows that compared to the climatology (i.e., the month-specific mean for each variable), the out-of-sample errors of the GloSea5 system are considerably low.

## Seasonal climate hindcasts

Hindcast data (i.e., historical forecasts) from the GloSea5 system [66,67] were retrieved using the Copernicus Climate Data Store [57] at monthly time steps for each of the 28 ensemble members, for lead times of 1 to 6 months ahead, and for the period January 2007 to December 2016. At the time of the computations, data for the period May to October 2016 were unavailable from the Copernicus Climate Data Store.

## Model specification

Let $Y_{i,t}$ be the number of dengue cases for province $i = 1, \cdots, I$ and time $t = 1, \cdots, T$ where $I$ is the total number of provinces in the data set, and $T$ is the total number of time steps for which the model is fitted using Bayesian generalised linear mixed models (GLMM). Models were fitted with a conditional negative binomial distribution given the number of dengue cases in the previous time step. The general algebraic definition of the models at the linear predictor scale is given by

$$log\,(\mu_{i,t}) = \alpha + log\,(P_{i,a[t]}) + \rho\,log\,(Y_{i,t-1} + 1) + \sum_k \beta_k X_{k,i,t} + \sum_j \epsilon_j L_{j,a[t]} + \gamma_{i,a[t]} + \delta_{i,m[t]} + u_i$$
$$+ v_i,$$
(3)

where $\alpha$ corresponds to the intercept; $log\,(P_{i,a[t]})$ denotes the logarithm of the population at risk for province $i$ and year $a[t]$, included as an offset to adjust case counts by population; $log\,(Y_{i,t-1}+1)$ denotes the logarithm of the observed number of dengue cases in the previous month plus one with an autoregressive parameter $\rho$; $X$ is a matrix of $k$ seasonal meteorological

explanatory variables with regression coefficients $\beta$; and $L_{j,a[t]}$ is a matrix of $j$ land cover (peri-urban and urban) variables with regression coefficients $\epsilon$. Long-term trends are modelled using province-specific unstructured random effects for each year ($\gamma_{i,a[t]}$). Seasonality is accounted for by using province-specific structured random effects for each calendar month ($\delta_{i,m[t]}$) with first order autoregressive prior to allow each month to depend on the previous one. Unknown confounding factors, such as interventions and spatial dependency structures representing, for example, human mobility, were incorporated using structured ($v_i$) and unstructured ($u_i$) random effects for each province $i$.

Spatial random effects were specified using a Besag–York–Mollie model [69], which incorporates a spatial effect with a Gaussian exchangeable prior to account for unstructured variation and a spatial effect with an intrinsic conditional autoregressive prior to account for spatially structured variability. Delayed effects of meteorological factors on dengue were accounted for by incorporating a 3-month moving average of temperature, precipitation, specific humidity, and DTR centred at lag of 1 month (i.e., each value is given the average of lag 0 to a lag of 2 months). Sea surface temperature anomalies in the Niño region 3.4 were lagged 0 to 3 months based on previous research [20,33,70] and exploratory analyses. No delayed effects were considered for wind speed based on previous studies, indicating a highest effect of wind at a lag of 0 months [71].

Flat priors were set to regression coefficients ($\alpha,\rho,\beta$), and penalising complexity priors were assumed for the precision for all random effects [72]. Priors were specified using a Gaussian distribution. We did not explicitly specify a prior distribution for the dispersion parameter ($\phi$). We used the default `pc.gamma` specification in the INLA R package [73], which assumes a Gamma distribution with values of $\phi$ within the interval 1–5668.2. Models were fitted in R version 3.6.1 using the INLA package [73]. The relevant R code is available at https://github.com/FelipeJColon/paper_dengue_superensemble.

## Model selection

The best subset of seasonal climate predictors leading to the lowest observed prediction discrepancies for a given model was obtained using an expanding window TSCV algorithm [54]. Land cover variables were included in all competing models as they varied annually and are unlikely to change significantly at monthly time steps. We iteratively fitted all possible models containing 1 seasonal climate predictor at the time, then 2 seasonal climate predictors, and so on, until all seasonal climate variables were included in a full model [20]. Thus, we tested 128 unique model specifications across 114 forecast months (i.e., January 2007 to December 2016). The predictive ability of each model was evaluated using the CRPS which generalises the mean absolute error for the case of probabilistic forecasts. The advantage of the CRPS compared to the mean absolute error (MAE) or root mean squared error (RMSE) is that the CPRS does not focus on a specific point of the probability distribution of the forecasts, but on their distribution as a whole. CRPS values were computed in R using the `SpecsVerification` package [74]. In response to peer review comments, we also evaluated the bias of the forecasts with the lowest CRPS to investigate whether models systematically over- or underpredicted dengue counts [75]. The bias for a continuous forecast at time $t$ was defined as [76]

$$B_t(P_t, x_t) = 1 - 2 \times (P_t(x_t)), \tag{4}$$

where $P_t$ is the empirical cumulative distribution function of the prediction for the true value $x_t$ [75]. Bias values range from −1 to 1, with $B_t = 0$ as an ideal value. The sharpness of the best performing models was assessed to investigate their ability to generate predictions within a narrow range [75]. The sharpness of forecasts at time $t$ was defined using the normalised

median absolute deviation about the median (MADN) [75,76]. We defined sharpness as

$$S_t(P_t) = \frac{1}{0.675}\, median(|y - median(y)|), \tag{5}$$

where $y$ is a forecast with cumulative distribution function $P_t$; the division by 0.675 ensures a value equivalent to the standard deviation if the predictive distribution is normal [75]. Sharpest models have $S_t = 0$, while blurred forecasts have $S_t \to \infty$ [75]. Bias and sharpness were calculated using the `scoringutils` [76] R package.

TSCV was implemented using an expanding window approach dividing the data set into multiple training and testing sets. The initial training set comprised data from August 2002 to December 2006. Each time step ($k$), a further month of data was added to the training set until the training set contained $n$-6 observations, where $n$ is the total number of observations in the set. The testing set comprised the climate hindcast data for the 6 months immediately after the last observation in the training set for each geographical area. Seasonal climate hindcast data were used to simulate the behaviour of an operational system over the period 2007 to 2016. We calculated the mean CRPS for each model specification across all time steps and ensemble members. The best 5 performing models were selected to create a superensemble.

### Prospective lagged dengue cases

The number of dengue cases occurring at time $t$ is directly dependent on the number of cases that occurred in the recent past. Previous research suggests that including the logarithm of the number of cases in the previous month $\log(Y_{t-1})$ helps accounting for such temporal correlation in disease transmission [77]. Additionally, incorporating $\log(Y_{t-1})$ as a covariate improves the predictive ability of seasonal climate-informed disease models by reducing residual dispersion [77]. One complication of accounting for the number of cases in the previous month in an operational system is that $\log(Y_{t-1})$ in the temporal window of the forecast can only be known up to time $t$+1. More specifically, to generate dengue forecasts 1 month ahead, $\log(Y_{t-1})$ for time $t$+1 corresponds to the logarithm of the number of dengue cases at time $t$. We used a GLMM with a negative binomial specification to estimate the $\log(Y_{i,t-1})$ for time $t$+1 to time $t$+6 to ensure that the number of cases in the previous month were exactly the same for all competing models. Thus, reconstructed lagged dengue time series was exactly the same for all models. The algebraic definition of the model is

$$log\,(\mu_{it}) = \alpha + log\,(P_{i,a[t]}) + \rho\,log\,(Y_{i,t-1} + 1) + \gamma_{i,a[t]} + \delta_{i,m[t]} + u_i + v_i, \tag{6}$$

with $\alpha$ as the intercept; $\log(P_{i,a[t]})$ as the population at risk in province $i$ at time $a[t]$, included as an offset; $\log(Y_{i,t-1}+1)$ is the logarithm of the observed dengue cases plus one in the previous month with regression coefficient $\rho$; $\gamma_{i,a[t]}$ as province-specific unstructured yearly random effect; $\delta_{i,m[t]}$ as province-specific structured monthly random effect with an AR1 autocorrelation term; and $u_i$ and $v_i$ as province-specific structured and unstructured random effects.

We then predicted the number of dengue cases for time $t$+1. The logarithm of the predicted number of cases at time $t$+1 was then used as $\log(Y_{i,t-1})$ for time $t$+2. We repeated these steps for each lead time in the forecast until time $t$+6. Note we assumed a conditional negative binomial distribution to generate one-step-ahead point predictions for the reconstructed lagged dengue time series. Once the reconstructed lagged dengue time series was complete, we fitted all competing models as indicated in the Model selection section.

## Baseline model

We developed a historical baseline model, which forecasts the same seasonal average of dengue incidence every month at each model run following [18,34,78] to compare all models to a common reference. The baseline model is specified as follows:

$$log\left(\mu_{it}\right) = \alpha + log\left(P_{i,a[t]}\right) + \delta_{i,m[t]} + u_i + v_i, \tag{7}$$

where $\alpha$ is the intercept; $log\left(P_{i,a[t]}\right)$ is the population at risk in province $i$ at time $a[t]$, included as an offset; $\delta_{i,m[t]}$ are province-specific monthly structured random effects with an AR1 autocorrelation term; and $u_i$ and $v_i$ as province-specific structured and unstructured random effects. We compared the predictive ability of the superensemble to that of the baseline model using the continuous rank probability skill score (CRPSS). The CRPSS is defined as follows:

$$CRPSS = 1 - \frac{CRPS_f}{CRPS_b}, \tag{8}$$

where $CRPS_f$ is the CRPS value of the forecast, and $CRPS_b$ is the corresponding CRPS value of the baseline model.

## Model superensemble

Given a number of competing models, we generated a model superensemble [38] stacking models according to their CRPS following suggestions from peer reviewers. Stacking involves averaging predictions from multiple models using weighted averages. We found model stacking outperformed Bayesian moving average. Therefore, this technique was adopted for the final version of the system. Stacking was performed in 3 stages [79]. On the first stage, we simulated 1,000 samples from the posterior distribution of dengue cases for each of the best performing models and the baseline model. Then, we calculated their CRPS for the historical period comparing the samples against observed dengue counts. In the second stage, optimal weights [80,81] were calculated for each model as follows:

$$w_i = \frac{(V_i)^{-1}}{\sum_{j=1}^n (V_j)^{-1}}, \tag{9}$$

where $V$ is the variance represented by the square of the CRPS of the $i^{th}$ forecast errors. A new set of weights were computed each time a forecast was issued using training data from all previous years. On the last stage, the 1,000 samples were combined using a weighted average. We then calculated the 2.5th, 50th, and 9.75th percentiles of their distribution.

## Outbreak detection evaluation

The predictive ability of the model for outbreak detection was evaluated using the Brier score [82]. For the Brier score, smaller values indicate better predictions. Four moving outbreak thresholds were defined to evaluate the predictive ability of the superensemble for outbreak prediction: (i) the endemic channel plus 1 standard deviation; (ii) the endemic channel plus 2 standard deviations; (iii) the 75th percentile of the distribution of dengue cases per month; and (iv) the 95th percentile of the distribution of dengue cases per month. The endemic channel was calculated as the number of dengue cases per month and per province over the previous 5 years in agreement with current practice at the Vietnamese Ministry of Health. The 75th and 95th percentiles were calculated over the whole observational period at each time step. We then calculated the probability of exceeding the moving outbreak threshold generating 1,000 samples from the posterior distribution of the point forecasts to reflect forecast uncertainty.

The Brier score was calculated as follows:

$$BS = \frac{1}{N}\sum_{t=1}^{N}(f_t - O_t)^2,$$ (10)

where $N$ is the number of predictions; $f_t$ is the forecast probability that an outbreak may happen; and $O_t$ takes the value of 1 if there was an outbreak or 0 if there was no outbreak. Brier scores were computed in R using the `scoring` package [83].

Public health officials may be more likely to take preventive action if the probability of observing an outbreak exceeds a certain value [18]. We undertook a signal detection analysis to determine the ability of the forecast probabilities to classify the predicted number of dengue cases as outbreaks. There are 4 possible outcomes in this analysis: hit (true positive), correct rejection (true negative), false alarm (false positive), and missed (false negative). A count for each of the 4 possible outcomes was produced.

### Relative value of using a forecasting system

Unlike other measures, the relative value ($V$) of the forecasts generated by the superensemble depends on requirements set by the user [84]. Typically, $V$ is evaluated in monetary terms and is particularly useful when the probability of occurrence of an adverse event (e.g., a major outbreak) is known. If the probability of occurrence of an outbreak is greater than the ratio of the cost of taking preventive action divided by the loss incurred by not taking action (C/L ratio) and an outbreak occurs, then it will pay off to take action. If the probability of occurrence of an outbreak is lower than the C/L ratio, then it does not pay off to take preventive action. If the probability is equal to the C/L ratio, it does not matter if action is taken or not. Our analysis provides an initial screening process to identify areas where a forecast is most likely to be cost-effective and thus, target of comprehensive economic evaluation studies in the future.

Given that data on the cost of taking preventive action and on the losses incurred by not taking action were unknown to us at the time of writing this manuscript, $V$ was estimated for a range of theoretical C/L ratios following [84] by comparing the mean cost of using the forecasting system for outbreak detection compared to the mean expense incurred by either never preventing outbreaks or, on the contrary, taking preventive action every month of the year. $V$ takes a value of 1 if the forecast is perfect and a value of 0 if it is no better than the default action plan. If $V$ is negative, it indicates that the forecast is so poor that it would be more cost effective not to use it. The algebraic definition of $V$ is

$$V = \frac{E(S) - E(A)}{E(S) - E(P)},$$ (11)

where $E(S)$ is the theoretical expense incurred by taking preventive action each month or the losses incurred by no taking action at all even when an outbreak occurred, whichever is the least expensive method when not using the forecast; $E(A)$ represents the total cost of the forecast calculated as the cost of type 1 (false positive) and type 2 (false negative) errors plus the cost of acting when an outbreak was predicted and it occurred (true positive); and $E(P)$ indicates the cost incurred with a perfect forecast where outputs are exclusively true positives and true negatives.

## Results

### Model selection

We fitted a total of 14,592 different models (128 unique model specifications across 114 forecast months, i.e., January 2007 to December 2016) by using all climate variable in isolation, as

well as all their possible combinations. One major issue with regression models is potential overfitting which may arise from data redundancy due to high correlation between predictors. The Spearman rank correlation coefficients calculated for all pairs of candidate predictors was less than 0.7, except for between minimum and maximum temperature ($\rho = 0.94$). However, these 2 variables were not present in any of the best performing models.

## Model superensemble evaluation

The 5 models with the lowest CRPS values (i.e., best performing) and the baseline model were used to generate a superensemble. It is noted that the 5 best performing models were the only models with a CRPS below 90 when computing prospective predictions. For CRPS, lower values indicate a smaller difference between the forecasts and the observations. There was strong seasonal variation in the weights assigned to each model (S3 Fig). All competing models had similar weights across the whole evaluation period (January 2007 to December 2016), whereas the baseline model had the lowest weights. We note that the weights assigned to the baseline model gradually decreased with time.

Averaged across all lead times, the superensemble led to slightly lower CRPS values than each of the competing models, including the baseline (Table 1). However, when stratified by lead time, the superensemble outperformed the competing models at all time leads but only outperformed the baseline at leads of 1 to 3 months (Fig 1A). The uncertainty of the forecasts increased with lead time. This is also evident when using the CRPSS for which values above 0 (indicating better predictions) were observed for the superensemble only for leads 1 to 3 months (Fig 1B). The CRPSS can be interpreted as the relative improvement observed from using a model compared to a reference baseline. Thus, a value of 0.4 would suggest that the values from a given model have an absolute error that is 40% smaller than the absolute error of the baseline model. The CRPSS indicates that at a lead time of 1 month, the superensemble and the competing models outperform the baseline by about 42%, whereas at a time lead of 6 months, their absolute error is 12.5% larger than that of the baseline model.

All models showed a tendency to overestimate the predicted number of dengue cases indicated by bias values above 0 (Table 1). Bias also increased as the forecast horizon increased, suggesting that the tendency to overestimate increases as the forecast horizon increases (Fig 1C). The model with the smallest bias was the baseline followed by the superensemble. The sharpness of the forecasts deteriorates with lead time (Fig 1D) as smaller values indicate a better forecast. The superensemble showed slightly worse sharpness than 2 of the competing

**Table 1. Verification metrics and seasonal climate predictors of the model superensemble and the best performing models.**

| Model | CRPS | Bias | Sharpness | Seasonal climate predictors |
|---|---|---|---|---|
| Baseline | 79.9 | 0.37 | 0.01 | None |
| Superensemble | 73.4 | 0.44 | 1.90 | All included in Models 1–5 |
| 1 | 79.8 | 0.49 | 3.12 | $SH_{02}$, $WS_0$, $DTR_{02}$, $SST_{03}$ |
| 2 | 80.2 | 0.49 | 1.67 | $SH_{02}$, $DTR_{02}$ |
| 3 | 80.9 | 0.48 | 1.69 | $SH_{02}$, $WS_0$ |
| 4 | 81.4 | 0.50 | 2.79 | $SH_{02}$, $DTR_{02}$, $SST_{03}$ |
| 5 | 81.5 | 0.50 | 2.67 | $SH_{02}$, $WS_0$, $SST_{03}$ |

$DTR_{02}$, diurnal temperature range averaged over a 3-month period lagged 0 to 2 months (˚C); $SH_{02}$, specific humidity averaged over a 3-month period lagged 0 to 2 months (dimensionless); $WS_0$, wind speed in the current month (m s$^{-1}$); $SST_{03}$, sea surface temperature anomalies in the Niño region 3.4 averaged over a 4-month period lagged 0 to 3 months (˚C). CRPS, continuous rank probability score. CRPS and sharpness assume values between 0 and infinity with an ideal value of 0. Bias assumes values between −1 and 1, with 0 as ideal.

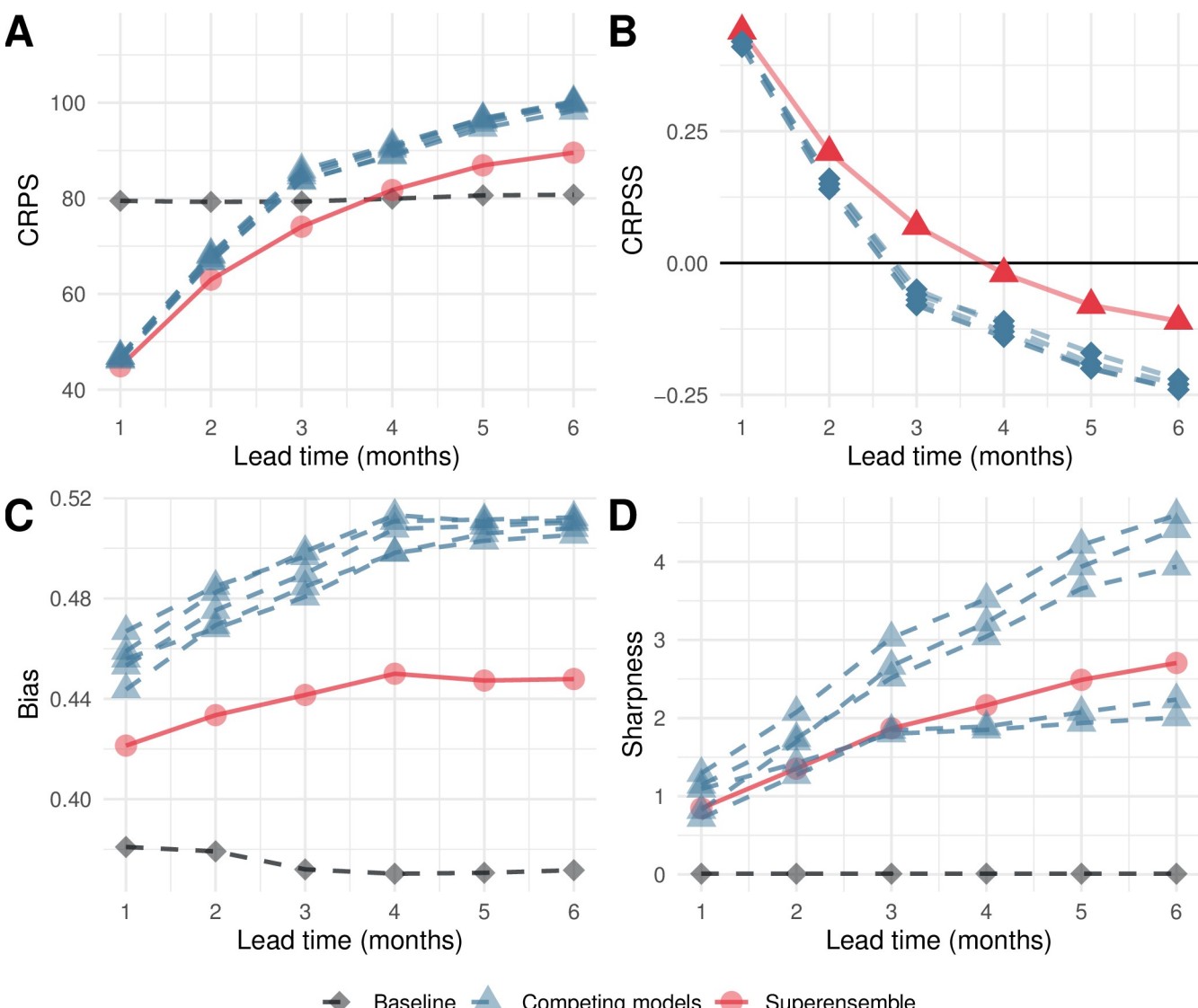

**Fig 1. Verification metrics by lead time.** Variation across all lead times averaged across the whole of Vietnam for (A) the CRPS, (B) the CRPSS, (C) the bias of the forecasts, and (D) the sharpness of the forecasts. Red lines indicate the performance of the model superensemble; blue lines depict the metrics for each of the 5 competing models; and grey lines indicate the behaviour of the baseline model. CRPS and sharpness assume values between 0 and infinity, with 0 representing a perfect forecast. Bias assumes values between −1 and 1, with 0 representing unbiased forecasts. CRPS, continuous rank probability score; CRPSS, continuous rank probability skill score.

models for lead times of 4 to 6 months. It is noted that the sharpness of the baseline model was always 0 as it generates the same seasonal values for all years in the data set (Table 1).

Across all metrics, the predictive ability of the superensemble deteriorated as the forecast horizon increased from 1 to 6 months. This situation is also evident where the forecast ensemble mean and its 95% credible interval (i.e., the interval in the domain of the posterior probability distribution) are plotted against the observed number of dengue cases (S4 Fig). Notice that the accuracy of the predictions worsens as the forecast horizon expands. We noted that the credible intervals of the predictions gradually became narrower as the number of months used to train the models increased (S4 Fig).

The predictive ability of the superensemble also varied with the month of the year. Overall, better performance was observed over the period July to December for CRPS, CRPSS, and bias

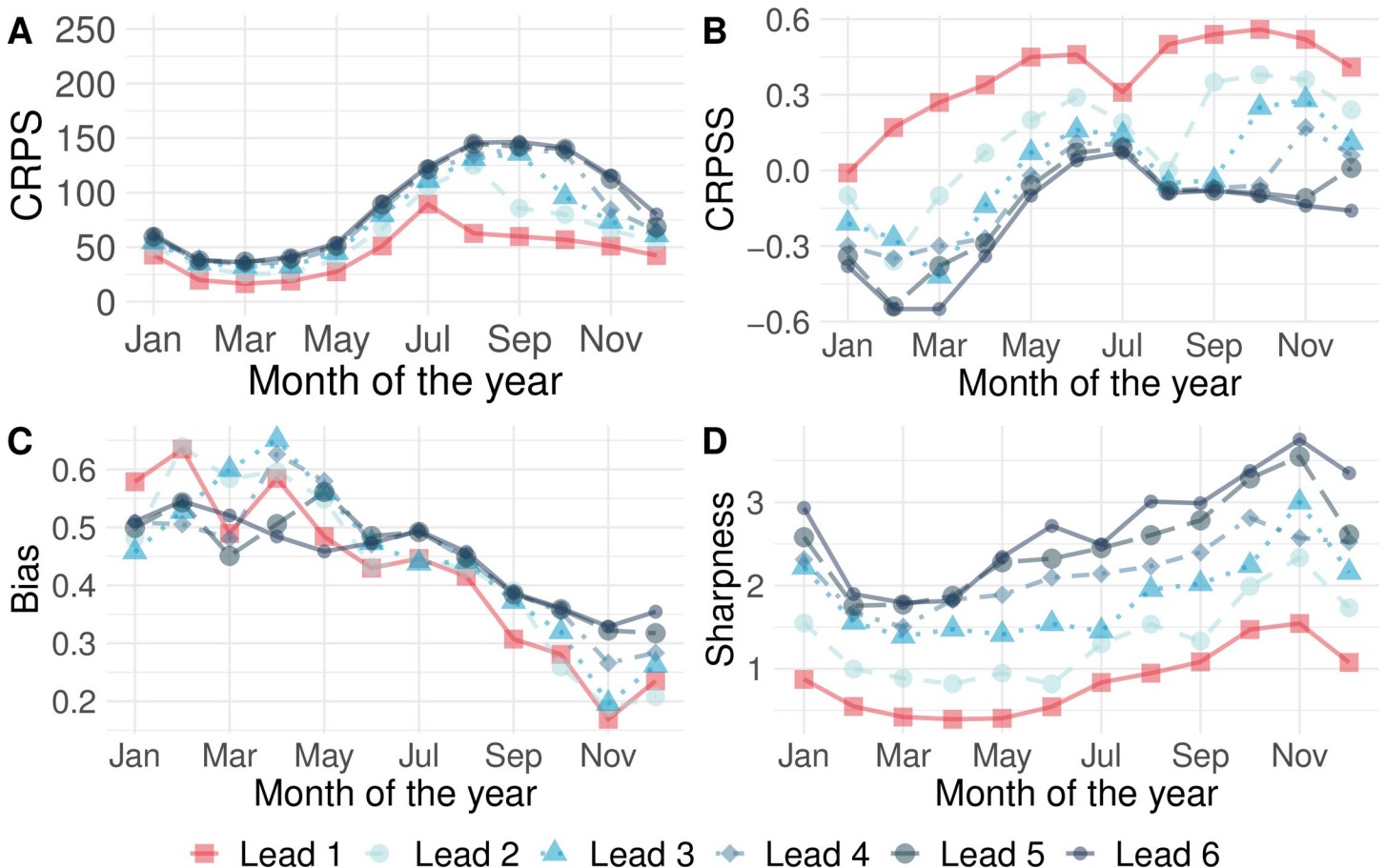

**Fig 2. Verification metrics by month of the year.** Variation across the months of the year averaged across the whole of Vietnam for (A) the CRPS, (B) the CRPSS, (C) the bias of the forecasts, and (D) the sharpness of the forecasts. Red lines indicate the performance of the model superensemble; blue lines depict the metrics for each of the 5 competing models; and grey lines indicate the behaviour of the baseline model. CRPS and sharpness assume values between 0 and infinity with an ideal value of 0. Bias assumes values between −1 and 1, with 0 as ideal. CRPS, continuous rank probability score; CRPSS, continuous rank probability skill score.

(Fig 2A–2C). The sharpness of the forecasts, however, deteriorated over the same period (Fig 2D).

Compared to the baseline model, the predictive ability of the forecast evaluated using the CRPSS showed significant spatiotemporal variation. Fig 3 shows that the CRPSS was consistently better than the baseline (red shaded areas) across most of the country for the period July to January. From February to June, however, the predictive ability of the baseline model is better than that of the superensemble for multiple provinces identified by CRPSS values below 0 particularly during March and April.

## Outbreak detection

Predictive ability was also evaluated by comparing the predicted probability of predicting outbreaks using the Brier score and 4 moving outbreak thresholds. Observed outbreak months were defined as months where the number of predicted dengue cases exceeded outbreak thresholds. As expected, highest predictive ability was achieved at a lead time of 1 month, after which the predictive ability of the superensemble gradually declined (Fig 4A). Across the forecast horizon, the highest predictive ability was observed when using an outbreak threshold based on the endemic channel plus 2 standard deviations, closely followed by the 95th

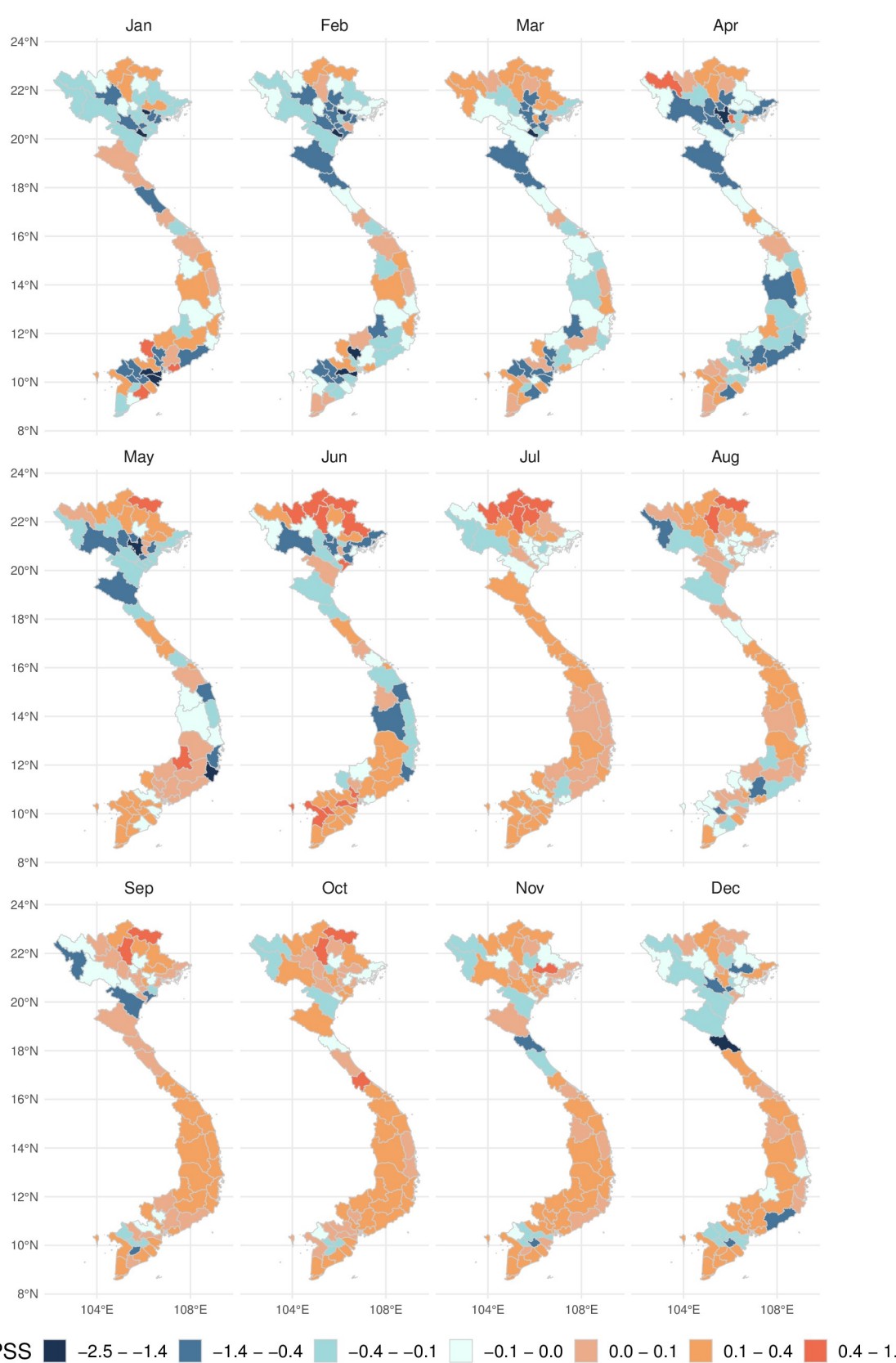

**Fig 3. Predictive ability across time and space.** Spatiotemporal variation of the CRPSS of the model superensemble. Orange shaded areas indicate a better performance of the superensemble compared to a baseline model. Blue areas indicate a lower performance of the superensemble compared to a baseline model. The shapefile used to create this figure was obtained from DIVA-gis (https://www.diva-gis.org). CRPSS, continuous rank probability skill score.

percentile. Stratified by month of the year, the predictive ability of the superensemble was generally greater between April and September for all thresholds except for the 75th percentile. When stratified by month, predictive ability was also larger using the endemic channel plus 2 standard deviations closely followed by the 95th percentile of the distribution of dengue cases.

There was significant spatial variation in the outbreak detection ability of the superensemble. Fig 5 shows that across all moving outbreak thresholds, outbreak detection ability was greater in the northern provinces compared to the central and southern provinces. It is noted that the 95th percentile threshold results in a larger number of provinces with low Brier scores (i.e., lower than 0.1) than the endemic channel plus 2 standard deviations.

A signal detection analysis indicates that using the superensemble with a moving outbreak threshold based on the endemic channel plus 2 standard deviations resulted in 73% correct predictions. Similarly, using the superensemble with an outbreak threshold based on the 95th percentile led to 72.5% correct predictions. The baseline model had the lowest predictive ability, with 61.3% correct predictions (Fig 6). However, using the 95th percentile outbreak threshold resulted in a slightly higher probability of detection (70% of all observed outbreaks) than using the endemic channel plus 2 standard deviations (68%). The baseline model had the lowest probability of detecting outbreaks (54.5%).

### Decision-support tools

**Portraying prospective forecasts.** The model superensemble was driven by seasonal climate forecasts to generate dengue forecasts for the period May to October 2020 using near-real time seasonal climate forecast data from the UK Met Office Global Seasonal forecasting system version 5 (GloSea 5) [66,67]. For the historical period (i.e., August 2002 to April 2020),

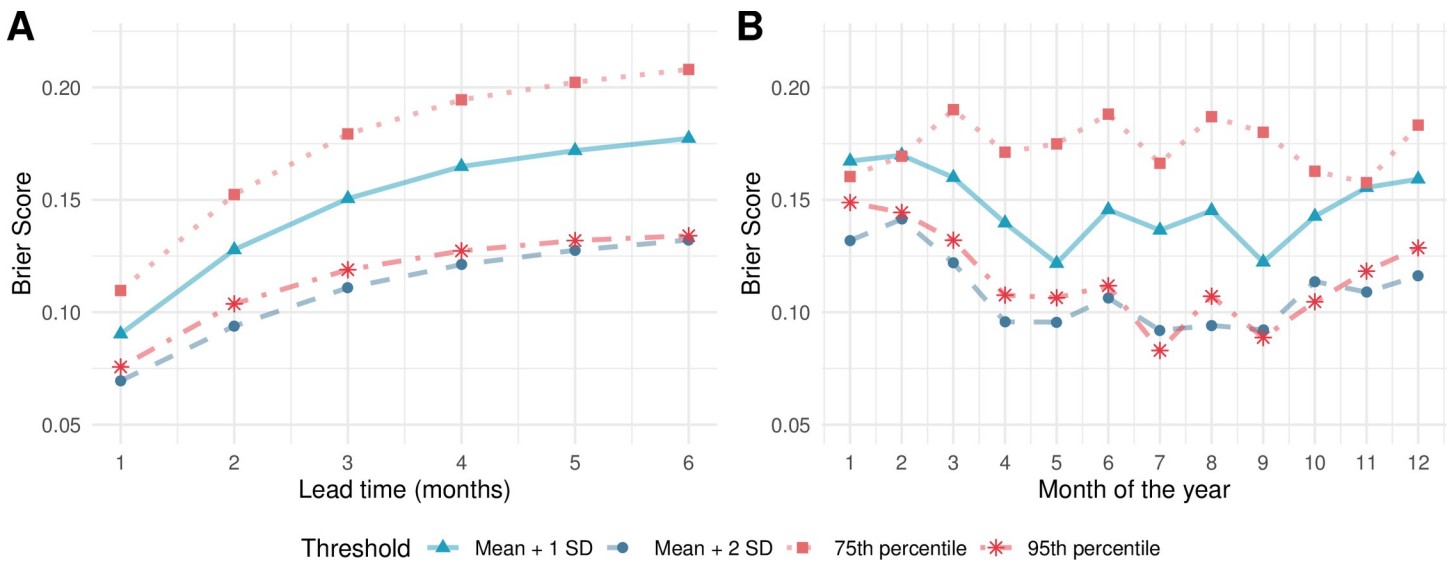

**Fig 4. Temporal variation of the Brier score.** Temporal variation of the Brier score averaged across all 63 Vietnamese provinces stratified by (A) lead time and (B) month of the year. The red lines indicate variation in the Brier score using percentile-related moving outbreak thresholds. The blue lines indicate variation in the Brier score using the endemic channel–related thresholds. The Brier scores assume values between 0 and 1, with 0 as ideal.

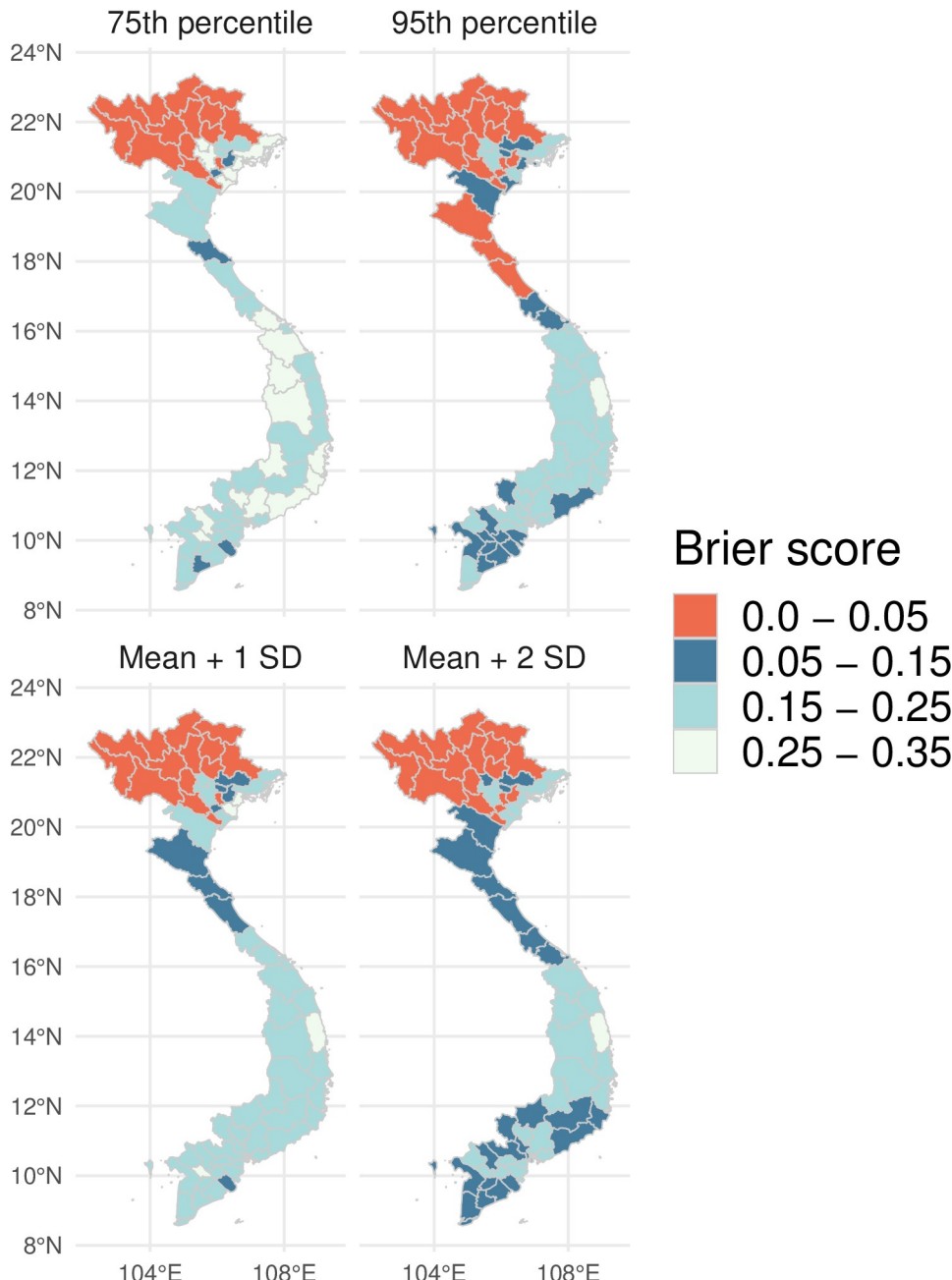

**Fig 5. Spatial variation in the Brier score.** Spatial variation of the mean Brier score averaged across all lead times calculated for 4 different moving outbreak thresholds over the period January 2007 to December 2016 and for each of the 63 Vietnamese provinces. Lower Brier scores (in orange) indicate a greater accuracy for detecting outbreaks. The Brier scores assume values between 0 and 1, with 0 as ideal. The shapefile used to create this figure was obtained from DIVA-gis (https://www.diva-gis.org).

the superensemble made slightly more accurate predictions (CRPS = 66.8, 95% CI 60.6 to 148.0) than a baseline model which forecasts the same incidence rate every month (CRPS = 79.4, 95% CI 78.5 to 80.5) at lead times of 1 to 3 months, albeit with larger uncertainty. For the period May to October 2020, the posterior median of the predictions made with the superensemble was also slightly more accurate (CRPS = 110, 95% CI 102 to 575) than the posterior

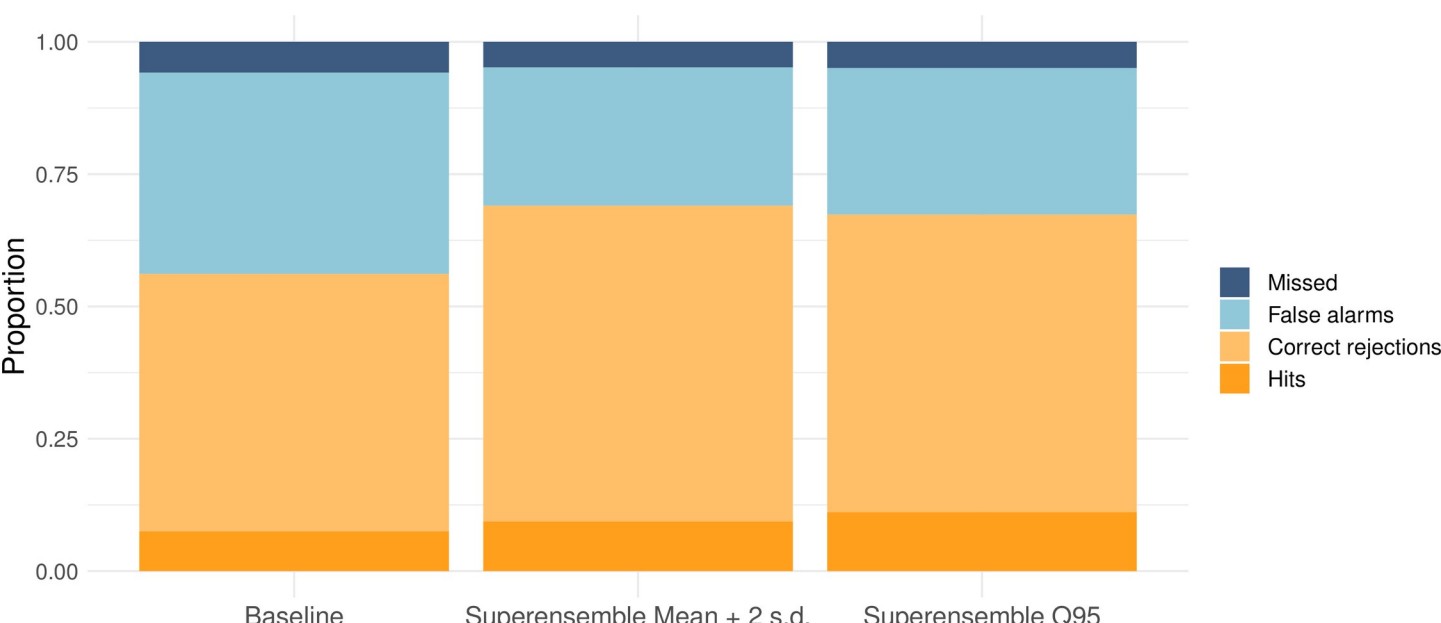

**Fig 6. Signal detection analysis.** Proportion of hits (dark orange), correct rejections (light orange), false alarms (light blue), and missed outbreaks (dark blue) for the baseline model and the model superensemble. Similar results were obtained at all lead times.

median of the predictions made with the baseline model (CRPS = 125, 95% CI 120 to 168), albeit with larger uncertainty. Fig 7 shows an example of how this information could be portrayed to users. In the figure, solid black lines indicate the posterior mean of the predicted values for each ensemble member. The red lines depict the observed dengue cases which were not known to the model at the time of the computation and are included here as a reference. Dotted black lines indicate the 95% credible interval of the predictions. Shaded areas indicate the low risk, moderate risk, high risk, and very high risk areas based on the endemic channel calculated using historical data from the previous 5 years (Table 2).

S5 Fig shows the posterior mean of the predicted values for each ensemble member and for 4 Vietnamese provinces where the system is being piloted along with their corresponding 95% credible interval (dashed lines). It can be observed that there is little spread between the 42 ensemble members indicating little between-member variability. As expected, the credible intervals increase as the forecast horizon increases, reflecting the uncertainties associated with the seasonal climate models used to generate the forecasts. It is noted that for none of the 4 provinces, the predicted mean number of dengue cases is above the high-risk region, suggesting that the September to February transmission season may be at normal conditions.

The spatial distribution of the posterior mean of the Model predictions could also be portrayed using risk maps (Fig 8) so that users can be made aware of the high-risk areas. Maps indicating the probability of exceeding a predefined outbreak threshold could also be easily created using the system outputs.

**Relative value.** An analysis of the relative value of using dengue forecasts generated by the superensemble can be undertaken using the model outputs to guide decisions-making processes. For such analysis, a value index ranging between 0 and 1 is used, with 1 indicating a perfect forecast [85,84]. The relative value of the forecasts is interpreted in comparison to either never taking preventative action or always taking action.

We defined a range of theoretical epidemic thresholds ranging between the 51st and the 99th percentiles of the distribution of dengue cases for the whole time series. Outbreak

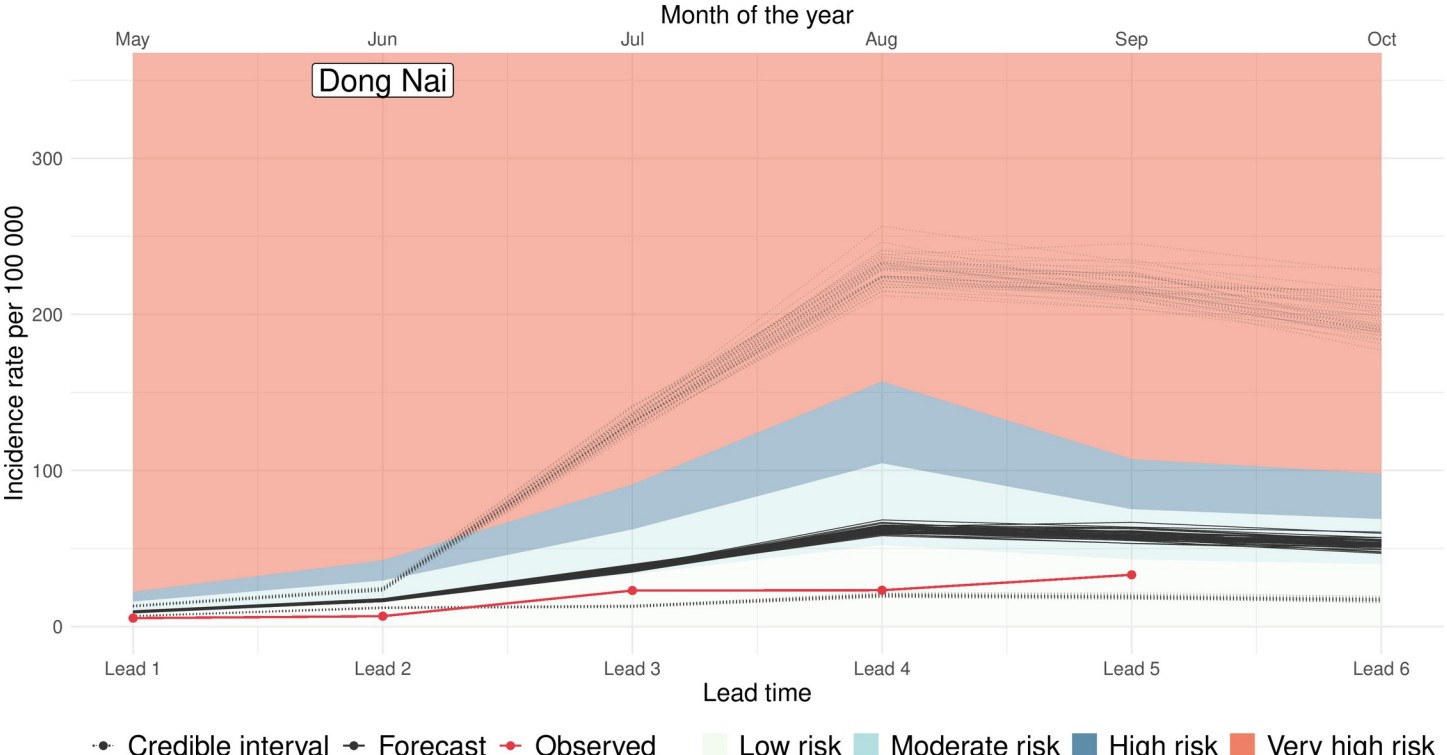

**Fig 7. Portraying prospective predictions.** Predicted dengue incidence rate for the period May to October 2020 for the Vietnamese province of Dong Nai using a model superensemble. The forecast was issued on May 10, 2020. The *x* axis indicates the time lead of the predictions. The *y* axis indicates the predicted dengue incidence rate. Solid black lines indicate the posterior mean estimate for each of the 42 forecast ensemble members. Red lines indicate the observed dengue incidence rate which was not known by the model at the time of the computation. The black dashed lines indicate the upper and lower bounds of the 95% credible intervals for each of the 42 ensemble members. The upper bound of the shaded areas indicate the month-specific risk based on the endemic channel plus 2 standard deviations calculated with historical data from the previous 5 years.

thresholds were province and month specific. Overall, the forecasts generated by the superensemble showed relative value in 76% of the provinces 1 month ahead, 73% of the provinces 2 months ahead, 68% of them 3 months ahead, and 65% of the provinces 4 to 6 months ahead (Fig 9). We note that the superensemble has relative value in areas where dengue is typically endemic such as the central and southern provinces. The superensemble had no relative value in the northern provinces where dengue is typically absent.

## Discussion

This study details a probabilistic dengue early warning system based on a model superensemble, formulated using Earth observations and driven by seasonal climate forecasts. The system

**Table 2. Thresholds used to define low, moderate, high, and very high risk levels based on an endemic channel.**

| Risk level | Endemic channel |
|---|---|
| Low | 0 to mean |
| Moderate | Mean to mean + 1 SD |
| High | Mean + 1 SD to mean + 2 SD |
| Very high | Larger than mean + 2 SD |

Mean, the mean number of dengue cases in the previous 5 years; SD, standard deviation of the observed number of dengue cases in the previous 5 years.

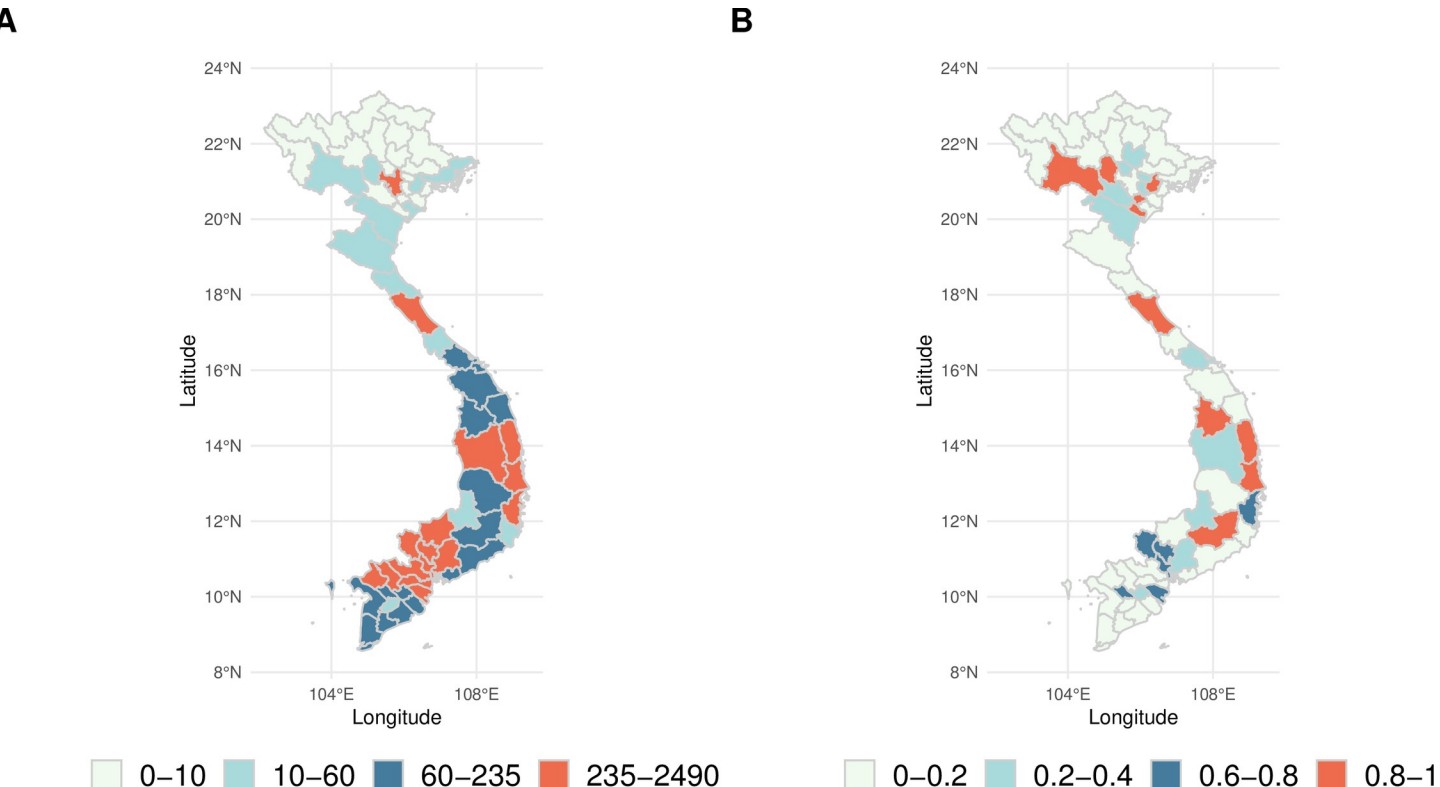

**Fig 8. Risk maps.** (A) Spatial distribution of the posterior mean of the predicted number of dengue cases 1 month ahead for a forecast initialised on May 10, 2020. (B) Spatial distribution of the probability of exceeding an outbreak threshold based on the 95th percentile. The shapefile used to create this figure was obtained from DIVA-gis (https://www.diva-gis.org).

is able to generate accurate probabilistic forecasts of dengue metrics that have the potential for guiding policy- and decision-making processes in Vietnam. We make predictions for each of the 28 ensemble members of the seasonal climate forecast and generate 1,000 samples from the posterior distribution of the predicted number of cases.

Deciding which predictive model is the best from a suite of competing models is not a straightforward task. Each model carries somewhat different information of the modelled processes. Here, we present a method for reconciling between-model disagreements while improving forecast accuracy. The combination of models into a superensemble helps offset individual model biases across time and space [86]. Superensembles were initially developed for climate modelling [87] and have gradually gained popularity in disease modelling (see, for example, [38,45,88]).

Our novel dengue early warning system relies on probabilistic models to reflect forecast uncertainty and to explicitly assign probabilities to outcomes [89]. The system has been developed to aid policy- and decision-making processes in Vietnam with the guidance of key stakeholders in the Ministry of Health of Vietnam, WHO regional office, the Pasteur Institutes in Nha Trang and Ho Chi Minh, Province-level Ministries of Health, the Vietnamese NIHE, and the Tay Nguyen Institute of Hygiene and Epidemiology. A range of stakeholder engagement workshops, face-to-face meetings with users, and surveys were conducted to tailor the system to the users' needs. Our results demonstrate that the forecasts generated by our spatio-temporal superensemble have the potential to guide changes to the current practice in dengue control towards a more preventative approach allowing bespoke and targeted public health

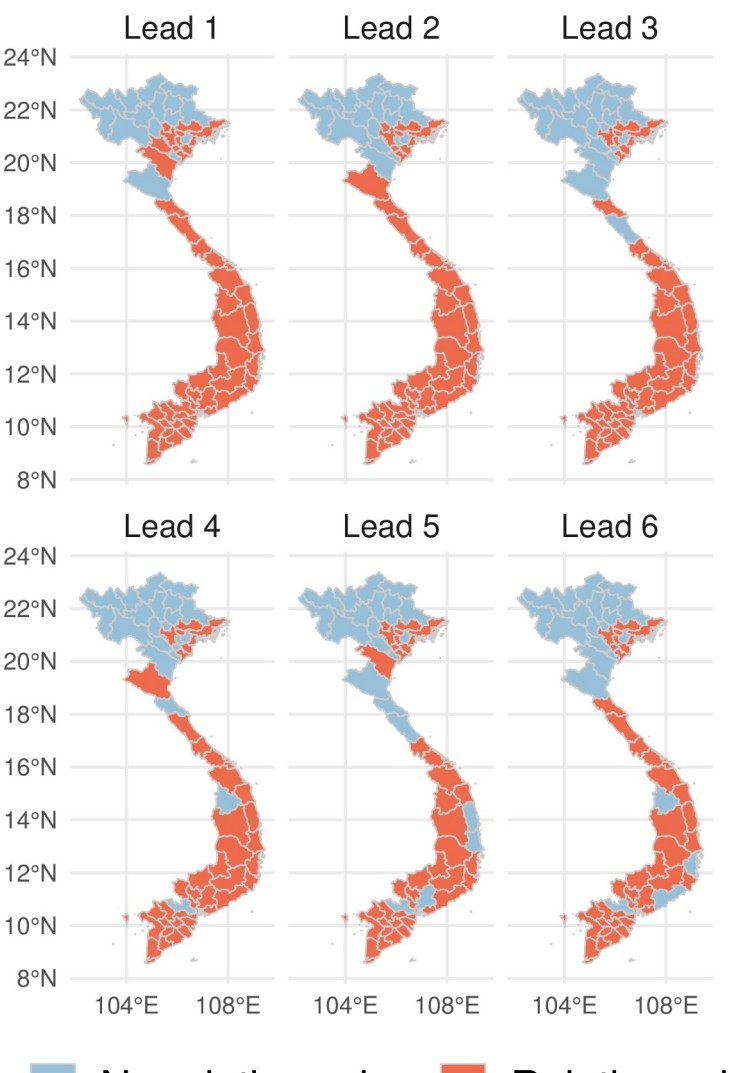

**Fig 9. Spatial variation of the relative value of the forecasts.** Spatial variation of the relative value of the forecasts generated by the model superensemble 1 to 6 months ahead. Orange shaded areas indicate provinces where there is relative value (based on a range of theoretical cost–loss ratios and outbreak thresholds). Blue shaded areas indicate provinces where the superensemble had no relative value. The shapefile used to create this figure was obtained from DIVA-gis (https://www.diva-gis.org).

interventions and a more efficient allocation of scarce resources. Research on use of the system by stakeholders and its effect on changing dengue control practices is currently ongoing.

We demonstrate that the superensemble slightly outperforms the predictive ability of the individual probabilistic models selected from a suite of top performing models at multiple lead times and months of the year in line with previous research [38,45,87,88]. Compared to a baseline model that predicts the same dengue incidence rate for each season and province, the superensemble generated more accurate predictions at lead times of 1 to 3 months but not beyond. Model performance was assessed using a range of verification metrics for probabilistic forecasts across time and space. To our knowledge, this is one of the first early warning systems informed by Earth observations to demonstrate predictive ability for prospective year-round

dengue prediction in a robust out-of-sample framework. It is also one of the first prototypes for routine dengue early warning at multiple time leads.

We found that the performance of the superensemble varied with geographic location, forecast horizon, and time of the year. The system showed ability in predicting spatiotemporal variations in dengue cases and outbreak occurrence at forecast horizons of up to 6 months ahead. Predictions deteriorated, and uncertainty increased with lead time, as previously observed in other settings and diseases [36,42,75,90]. Forecasts improved, and the credible intervals decreased as time progressed and dengue data increased, likely due to an improvement in the associations learned by the superensemble. Forecast errors increased for the onset and peak of the transmission season, possibly due to substantial interannual variation (S6 Fig).

Relative to a baseline model, the superensemble made slightly more accurate predictions of the estimated number of dengue cases across most provinces, at lead times of 1 to 3 months in agreement with previous studies [18,42,45]. At lead times beyond 3 months, however, the baseline model made more accurate predictions possibly because, as suggested by [45], incorporating seasonal climate data into predictive models may increase model complexity at the expense of lower out-of-sample predictive ability. Previous research has found that climate naïve models may outperform climate-informed models [32,45]. Those studies, however, used different sources of climate and environmental data than our study and were conducted in different settings (i.e., Thailand, Peru, and Puerto Rico) where the effects of climate might be significantly different to those observed in Vietnam. In addition, those studies initialised their prospective forecasts at fixed times of the year, whereas our study initialised forecasts at each calendar month, providing the model with more opportunities to learn the relationships between dengue incidence and each of the climatic and sociodemographic predictors.

Outbreaks are difficult to predict, even more so at forecast horizons of several months ahead. Nevertheless, our superensemble demonstrated predictive ability for outbreak detection up to a lead time of 6 months, evaluated using proper scores over a suite of moving outbreak thresholds. One of the best performing outbreak thresholds was the endemic channel plus 2 standard deviations, computed using data from the previous 5 years. This method is currently in use across Vietnam [48,49], although with the difference of excluding outbreak years from the computation of the threshold. Following our results, however, we recommend the use of the 95th percentile as a threshold for outbreak detection as it gave slightly better results for detecting outbreaks. Recognising the limitations of the province-level data, it is encouraging that our predictions are accurate in most provinces up to 6 months in advance.

In disease forecasting, each decision (e.g., to prevent or not to prevent an outbreak) has an associated cost that will lead to a benefit or a loss depending on the outcome. Decision-makers have the task of selecting the action that minimises potential losses. We used a simple analysis [84] to assess the relative value of the forecasts generated by the superensemble. Our figures are only illustrative. Still, they highlight that using a dengue early warning system has relative value compared to not using a forecast. The superensemble had considerable value across most provinces. However, in northern provinces, where dengue is essentially absent, the forecast is predicted to have limited relative value compared to always preparing for an outbreak or never preparing for it. The assessment of the relative value of the dengue early warning system may help stakeholders justify public investment in the development and generation of seasonal forecasts or to help raise awareness of their potential value.

Although our proposed early warning system provides useful information for public health preparedness and response, it has some limitations worth mentioning. First, while our modelling framework incorporates important determinants of dengue occurrence such as climate and urbanisation, it does not explicitly incorporate, at this stage, the potential effects of other important determinants of disease such as the deployment of mosquito control interventions,

vector indices, serotype-specific circulation, herd immunity, and the mobility of people and goods, all of which may lead to significant changes in the level of risk experienced locally [91]. Stakeholders in Vietnam have recently started collecting supporting data on vector indices and dengue virus serotypes. However, there is a lack of publicly available, continuous, and long-term data sets that could be used to inform modelling efforts. In this study, we account for some of the variation that might be attributed to these factors by using spatiotemporal random effects in each of the models included in the superensemble. Future developments of the system may incorporate some of these factors if data are made available. Second, the quality and consistency of the dengue data are affected by the limited confirmation of suspected cases through laboratory diagnostic test, leading to large uncertainties that are difficult to quantify. Third, our computations of dengue risk do not take into consideration uncertainties due to the potential under- or misreporting of dengue cases. Consequently, our model superensemble forecasts may underestimate the real number of cases occurring at any given time. Finally, our forecasts do not carry forward the uncertainty of the one-step-ahead forecasts used to simulate prospective lagged cases. Future developments may represent this uncertainty more accurately by using simulation-based path forecasts in a Bayesian framework [92].

## Conclusions

We have demonstrated that a superensemble of probabilistic dengue models, formulated using Earth observations and driven by seasonal climate forecasts and a reconstructed/prospective estimate of lagged dengue cases, is useful to generate probabilistic predictions of dengue risk across Vietnam at multiple lead times and months of the year. We acknowledge that there are alternative state-of-the-art approaches to dengue forecasting that may be equally as effective as the one presented here. However, our aim was to develop a system that outperformed previous practice in Vietnam where no forecasting system was used to guide decision-making processes. A theoretical analysis showed that compared to not using a forecasting system, the superensemble has relative value across most of the country, suggesting that it has potential for guiding decision-making processes. The dengue forecasting system presented here has been rolled out across Vietnam and could be tailored for other dengue-endemic countries. Further work may include investigating the feasibility of producing probabilistic forecasts with sufficient predictive ability at the district or commune levels and a comparison with other statistical and mechanistic approaches.

## Supporting information

**S1 Text. Prospective analysis plan.**
(DOCX)

**S1 Fig. Observed dengue cases across Vietnam.** Time series of monthly dengue cases from the 63 provinces in Vietnam (August 2002 to March 2020). Provinces are ordered from north (top) to south (bottom) according to the latitude coordinates of their centroid. White boxes indicate missing data.
(TIF)

**S2 Fig. Verification metrics of the seasonal climate forecasts evaluated across Vietnam.** The *x* axis indicates the month of the year. The *y* axis indicates value of the CRPS for each variable. The lines indicate the lead time for the forecast. CRPS, continuous rank probability score.
(TIF)

**S3 Fig. Optimal weights.** Seasonal variation of the optimal weights assigned to each of the models included in the superensemble over the period January 2007 to December 2016. (TIF)

**S4 Fig. Observed vs. predicted dengue cases.** Observed (dashed lines) and predicted (solid lines) dengue cases across Vietnam aggregated at the national level. Shaded areas represent the 95% credible interval. Predictions are shown for the forecast horizons of 1 (top), 3 (middle), and 6 (bottom) months ahead. Data correspond to the period January 2007 to December 2016. (TIF)

**S5 Fig. Prospective dengue predictions.** Predicted dengue cases for the period May to October 2020 for 4 pilot Vietnamese provinces using a model superensemble. The forecast was issued on May 10, 2020. The $x$ axis (top) indicates the month of the predictions. The $x$ axis (bottom) indicates the time lead of the predictions. The $y$ axis indicates the predicted incidence rate. Black solid lines indicate the posterior mean estimate for each of 42 forecast ensemble members. Red lines indicate the observed dengue incidence rate which was not known by the model at the time of the computation and are included here as a reference. The black dashed lines indicate the upper and lower bounds of the 95% credible intervals for the 42 ensemble members. The upper bound of the shaded areas indicates the month- and province-specific percentiles based on dengue data for previous 5 years. (TIF)

**S6 Fig. Month-specific variability in dengue cases across Vietnam.** The $x$ axis indicates the month of the year. The $y$ axis indicates increases in the number of dengue cases (square root transformed). The upper and lower limits of each box represent the interquartile range of the distribution of dengue cases for each month. The middle solid line indicates the median value. The upper and lower whiskers indicate the maximum and minimum values of the dengue case distribution (excluding outliers which are indicated with dark purple circles). Outliers are values beyond 1.5 times the interquartile range. (TIF)

## Acknowledgments

We would like to thank Dr Satoko Otsu and Dr Trang Cong Dai from the World Health Organization Office in Vietnam and Mr Dao Khanh Tung from the United Nations Development Programme Office in Vietnam for the helpful discussions and support for the development of this project. We would also like to acknowledge the support and insights of the Pasteur Institute Ho Chi Minh City, the Pasteur Institute Nha Trang, the Institute of Hygiene and Epidemiology Tay Nguyen (TIHE), and the National Institute of Hygiene and Epidemiology (NIHE).

## Author Contributions

**Conceptualization:** Felipe J. Colón-González, Leonardo Soares Bastos, Oliver J. Brady, Rachel Lowe.

**Data curation:** Barbara Hofmann, Alison Hopkin, Quillon Harpham, Tom Crocker, Rosanna Amato, Iacopo Ferrario, Francesca Moschini, Samuel James, Sajni Malde, Eleanor Ainscoe, Vu Sinh Nam, Dang Quang Tan, Nguyen Duc Khoa, Mark Harrison.

**Formal analysis:** Felipe J. Colón-González, Leonardo Soares Bastos.

**Funding acquisition:** Mark Harrison, Gina Tsarouchi, Darren Lumbroso, Oliver J. Brady, Rachel Lowe.

**Investigation:** Felipe J. Colón-González.

**Methodology:** Felipe J. Colón-González, Leonardo Soares Bastos, Oliver J. Brady, Rachel Lowe.

**Project administration:** Gina Tsarouchi, Darren Lumbroso, Oliver J. Brady, Rachel Lowe.

**Resources:** Barbara Hofmann, Alison Hopkin, Quillon Harpham, Tom Crocker, Vu Sinh Nam, Nguyen Duc Khoa, Gina Tsarouchi, Darren Lumbroso, Oliver J. Brady, Rachel Lowe.

**Software:** Felipe J. Colón-González, Leonardo Soares Bastos, Oliver J. Brady, Rachel Lowe.

**Supervision:** Oliver J. Brady, Rachel Lowe.

**Validation:** Felipe J. Colón-González, Barbara Hofmann.

**Visualization:** Felipe J. Colón-González, Leonardo Soares Bastos.

**Writing – original draft:** Felipe J. Colón-González.

**Writing – review & editing:** Felipe J. Colón-González, Leonardo Soares Bastos, Barbara Hofmann, Alison Hopkin, Quillon Harpham, Tom Crocker, Rosanna Amato, Iacopo Ferrario, Francesca Moschini, Samuel James, Sajni Malde, Eleanor Ainscoe, Vu Sinh Nam, Dang Quang Tan, Nguyen Duc Khoa, Mark Harrison, Gina Tsarouchi, Darren Lumbroso, Oliver J. Brady, Rachel Lowe.

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
