## [Editor Report · Decision Letter 0]

29 Jun 2020

Dear Dr Colón-González, 

Thank you for submitting your manuscript entitled "Probabilistic seasonal dengue forecasting in Vietnam using superensembles" for consideration by PLOS Medicine.

Your manuscript has now been evaluated by the PLOS Medicine editorial staff and I am writing to let you know that we would like to send your submission out for external peer review.

Kind regards,

Artur Arikainen,

Associate Editor

PLOS Medicine

---

## [Decision Letter · Decision Letter 1]

5 Aug 2020

Dear Dr. Colón-González,

Thank you very much for submitting your manuscript "Probabilistic seasonal dengue forecasting in Vietnam using superensembles" (PMEDICINE-D-20-02989R1) for consideration at PLOS Medicine. 

Your paper was evaluated by a senior editor and discussed among all the editors here. It was also evaluated by five independent reviewers, whose comments are enclosed below; I hope you find the reviews constructive. Any accompanying reviewer attachments can be seen via the link below:

[LINK]

In light of these reviews, I am afraid that we will not be able to accept the manuscript for publication in the journal in its current form, but we would like to consider a revised version that addresses the reviewers' and editors' comments. Obviously we cannot make any decision about publication until we have seen the revised manuscript and your response, and we plan to seek re-review by one or more of the reviewers. 

We expect to receive your revised manuscript by Aug 26 2020 11:59PM. Please email us (plosmedicine@plos.org) if you have any questions or concerns.

We look forward to receiving your revised manuscript. 

Sincerely,

Emma Veitch, PhD

PLOS Medicine

On behalf of Clare Stone, PhD, Acting Chief Editor,

PLOS Medicine

plosmedicine.org

*Please revise your title according to PLOS Medicine's style - this should summarise the study question but also state (preferably after colon in the subtitle) the study design (eg "A randomized controlled trial," "A retrospective study," "A modelling study," etc.).

*In the last sentence of the Abstract Methods and Findings section, please add a brief summary of any key limitation(s) of the study's methodology.

*At this stage, we ask that you include a short, non-technical Author Summary of your research to make findings accessible to a wide audience that includes both scientists and non-scientists. The Author Summary should immediately follow the Abstract in your revised manuscript. This text is subject to editorial change and should be distinct from the scientific abstract. Please see our author guidelines for more information: https://journals.plos.org/plosmedicine/s/revising-your-manuscript#loc-author-summary

*Please clarify whether the analytical approach detailed here was laid out in a prospective protocol or analysis plan? Please state this (either way) early in the Methods section.

Comments from the reviewers:

Reviewer #1: Dengue is a climate-sensitive disease, and many efforts have been attempted to predict future outbreak risk using environmental conditions and past patterns of dengue. However, it is relatively scarce to combine forecasts from multiple competing models into a superensemble to make more accurate predictions. This study aimed to develop a superensemble of probabilistic hierarchical dengue models to predict dengue risk up to six months ahead at the province level across Vietnam. It is an interesting and well-presented manuscript but several issues should be addressed to improve its clarity. 

How were suspected and confirmed dengue cases diagnosed and reported? More information on each category is clearly needed. The quality and consistency of the dengue data was likely to vary with area and time, and it remains largely unknown how this problem might affect the construction of the model. Data on both disease and environmental conditions (e.g. meteorological, demographic and land-cover variables) had different spatial resolutions and it is unclear how these datasets were matched. Is there any change in the administrative boundary across 63 provinces in Vietnam during the period 2002-2019? How were population weighted monthly averages calculated for each climate variable? Why were verification metrics inconsistent (e.g. CRPS, RMSE and MAE were small but DIC and WAIC were big in Table 1)? Several explanatory variables (e.g. minimum temperature, maximum temperature, and sea surface temperature anomalies) were likely to be highly correlated (L315-319), and how was this issue resolved? Combining all best performing models into a model superensemble led to a lower CRPS and MAE than any of the five competing models, but the RMSE was an exception as the RMSE of the superensemble was higher than two of the five models (L335-337). Why is it the case? Fig 8 shows a range of cost-loss ratios and outbreak thresholds for 42 Vietnamese provinces. How about other 21 provinces as Vietnam has 63 provinces? It is important to demonstrate that their spatio-temporal superensemble could guide changes to the current practice (L479-84). However, no results were presented on this aspect.

Reviewer #2: I have mixed feelings about this paper. On the one hand, the work is really quite impressive, with a diverse set of top-notch datasets and cutting-edge methods. I found myself nodding away enthusiastically as I read the methods section and text of the results. On the other hand, though, the actual results are not very good. The superensemble of forecasts gives effectively the same performance as the better models do individually, as can be seen from table 1 but more clearly from supplementary figure 2. The text therefore feels misleading: yes, the superensemble may be technically better but effectively it was the same and I would have wanted to have seen a more balanced description of that, because this is the sort of project that other countries should be copying and they need the right information to evaluate how complex to aim for. The other reason I say the results are not very good is that the forecast at six months is really poor. Figure S6 illustrates this clearly: in March, the authors are trying to forecast the situation in August, and for HCMC can't even tell if the number of cases will be a few hundred or 20 000. If there is this level of uncertainty, the more honest thing to do would be to report that the forecast accuracy is intolerable at that forecast window and instead limit the forecast to say one to two months into the future. I thought this paper was great until I went to the supplement and had the chance to see directly what the forecasts looked like: they are really not presented transparently within the ms itself.

My second major concern is about the cost analysis, which is poorly handled and lowers the overall quality of the manuscript. I don't think the equation is correct (surely the optimal decision depends not just on the ratio of cost under action 1 and benefit under action 2 but also the impact of the two actions on the outcome and the costs of action 2 and benefits of action 1), and the statement about always or never preparing for outbreaks being the default raises alarm bells. The decision making process about interventions for outbreaks of dengue is waaaaaay more complex than the authors are making it out to be and the analysis here really comes across as the view from an ivory tower. If the authors really want to do a cost-benefit analysis, perhaps because the grant promised it, then it would behove them to do it properly. You have some really great people like Mark Jit at LSHTM, why not tap on them to see how to do a proper CBA? (Probably as a separate paper though.)

My final major criticism is on the structure of the forecast model/s themselves. Why are no lags considered for the predictors? Some such as El Nino would expect to see their presence felt a long time after. Even local features could plausibly have an impact longer than one month in the future because of the long latency caused by the cycles of breeding, infection and both incubation periods. You should be able to quickly refute this. Furthermore, why are the data at such a crude temporal resolution? If the VN MOH has provided an author, can s/he not provide weekly (or even daily! That would be fab!) data, because that would presumably greatly improve the predictive potential.

A few minor comments that may help revise the manuscript:

Abstract: "the skill of a system using different lead times in a year-round prediction system has not been previously explored" I'm not sure that is strictly true. For instance, Singapore has been doing something like this for a number of years 

L13: \\textit{Ae.~aegypti} to prevent inter-sentence spacing. You can similarly use this trick after i.e.~ and so on

L10: International standard is to use a space to demarcate thousands from hundreds to avoid ambiguity about decimal point vs decimal comma 

L24-32: A bit of repetitive sentence structuring, consider tweaking.

L39-48: These sentences were a bit hard to read. Consider tweaking with some extra words at the start of the sentences like "It is also known that", "In contrast," or changing the pattern a bit like "Biting rates are also affected by xxx" etc, to help with the flow

L42: bad latex: $>$ instead of > and either \\degree, \\textdegree (depending on packages) or $^{\\circ}$

L71: What is fundamentally wrong with testing a model on year 2015 after training it on 2016 (say)? Does it not still give a valid approximation of out of sample performance?

L81: Good point, perhaps worth elaborating on the difficulties of implementing such a system not in a research environment

L102: $<$ not <

L129: Possible to show time series of the data for a few provinces?

L142: Perhaps my physics is letting me down but isn't kg kg^{-1} unitless?

L152-160: Do I read correctly that you used two different datasets to create population data? How were these harmonised? If not, please clarify and remedy my confusion!

Eqn1: Should Y be Y_{i,t-1}? Should the sum be enclosed in parenthesis to prevent ambiguity about the random effects? What happens when Y=0?

L205: missing closing parenthesis

Eqn 5: \\exp \\max

T1: Interesting that so little benefit accrues from the superensemble compared to simpler models. The text hints that the superensemble is substantially better

L367: Some of this subsection should perhaps go in the methods section?

L407/8: missing spaces

L416: The proposed decision support tool is… a bit weak, isn't it? Aesthetically it's ugly, but the uncertainty in the predictions seemingly makes the forecast useless. 

T2: Why is Moderate defined as being up to the 75th percentile AND mean+1SD? Mean+1SD should not be the 75th percentile.

L515: What is 95 per cent percentile? Is it the same as the 95th percentile?

F1: Not very informative. Can I propose you omit the islands (and just add a comment in the legend that islands in the East Sea are omitted for space). Then you can stack the maps in a 6x2 arrangement, making them all bigger and reducing the white space between them. At present you can hardly make out the changes in colour around HCMC and Vinh.

Reviewer #3: # Overview

In this manuscript the authors present and evaluate ensemble forecasts of

monthly dengue case numbers in each province of Vietnam for 2002-2019, which

incorporate a variety of demographic, land-cover, and climate data. These

forecasts are evaluated for lead times of 1-6 months --- looking much further

ahead than many existing dengue forecasting studies. The evaluation includes a

cost-loss analysis of whether action taken in response to the projected outbreak

probability would be beneficial. I found this to be an interesting and

well-designed study, and the prospect of further evaluation of this forecasting

system is exciting.

# Major comments

1. Regarding the "Modelling approach" section on page 6, I understand that

 there were 7 seasonal climate data sets (mean monthly values were used):

 (a) Minimum temperature;

 (b) Maximum temperature;

 (c) Daily precipitation;

 (d) Specific humidity;

 (e) Diurnal temperature range;

 (f) Wind speed; and

 (g) Sea surface temperature anomalies for the Niño 3.4 Region.

 So 128 different models were considered to cover all possible combinations of

 these 7 data sets. But this only became apparent after reading the first few

 sentences of the results section, where this is explained to the reader. I

 would have found it quite helpful to instead include these remarks in the

 "Modelling approach" section.

2. "Model selection" section (page 6): "The testing set comprised the climate

 hindcast data for the six months immediately after the last observation in

 the training set for each geographical area."

 I apologize if I've missed an obvious point here, but why were hindcasts used

 instead of actual data for this period? Is the intent that in an operational

 system they would be replaced by climate forecasts from the same source

 (Copernicus Climate Data Store)?

3. "Accounting for autocorrelation in disease transmission" section (page 7):

 "To generate dengue forecasts one month ahead, log(Y_i, t−1) for time t + 1

 corresponds to the logarithm of the number of dengue cases at time t. To

 generate the forecasts two months ahead we first fitted a climate naïve model

 ..."

 It isn't clear to my why a climate-naive model was used here. Since the

 testing set included climate hindcast data for the six months after the last

 observation (see previous comment) then why couldn't the climate-informed

 model be used to predict the number of dengue cases for time t + 1? And at

 the start of the Results section (page 9), the authors write:

 "Predictions up to six months ahead were made at each time step starting in

 January 2007 using seasonal climate hindcasts (retrospective forecasts) data

 ..."

4. "Model superensemble" section (page 8): "A new set of superensemble weights

 were computed each time a forecast was issued using training data from all

 previous years."

 Can the authors comment on how these weights evolved over time? Was the same

 model consistently the highest-weighted over all of the forecast months?

 Presumably in an operation system it would be highly desirable if the model

 weights were stable from one month to the next? I suppose this could be

 extended to allowing seasonal variation (i.e., month-to-month) if different

 models are understood to perform best in different calendar months. It would

 be great to see, e.g., a supplementary figure that shows how the weights

 evolved.

4. "Model superensemble" section (page 8): "In addition, we evaluated the skill

 of the obtained forecasts using the continuous rank probability skill score

 (CRPSS). CRPSS is defined as: ..."

 It appears that there should be further text after equation (6), since it

 ends with a trailing comma, but instead the section ends. Is there missing

 text, or should the comma be removed?

5. Results section (page 9): how did the authors decide to include precisely 5

 models in the superensemble? Was there a clear divide in performance between

 the top 5 models and the remaining 123 models? And were the authors surprised

 that none of the models in the superensemble included daily precipitation as

 a seasonal climate predictor?

6. Results section (page 10): "The predictive ability of the [superensemble]

 model also varied with the month of the year (Figure 1). Overall, larger

 discrepancies between observed and predicted values were observed between

 July and December, when typically more cases are reported."

 Did the authors consider evaluating the models using any relative error

 measures, in addition to the chosen absolute error measures? This is an

 important consideration for infectious diseases epidemic forecasts since an

 error of, say, 10 cases has very different significance when the expected

 number is 20 cases or 20,000 cases. As the authors note:

 "From August to November the skill is reduced for selected provinces in the

 central and southern regions characterized by larger dengue incidence

 variability."

 This might be primarily influenced by the choice of absolute error measures

 rather than being intrinsic to the models themselves.

7. Results section (page 10): "S5 Fig. shows the observed and posterior

 predictive mean dengue cases across the 63 provinces computed one month ahead

 using the model superensemble. The model superensemble is able to reproduce

 the spatiotemporal dynamics of dengue fever with reasonable skill although

 predictions tend to underestimate the number of cases particularly during

 large outbreaks."

 This observation made me wonder whether the model fitting may not have

 selected appropriate values for the dispersion parameters.

 The authors note (top of page 6) that models were fitted "using a negative

 binomial specification to account for potential over-dispersion in the data"

 and that "penalising complexity priors were assumed for [...] the dispersion

 parameters".

 What prior distribution was used for the over-dispersion parameters, and what

 were the posterior distributions when the models were fitted? I couldn't see

 this mentioned in the manuscript text, nor am I able to identify a location

 in the provided code or supplementary materials where this is defined. In the

 provided file "04_Fit_models.R" there is the following code:

 ```

 mu1 <- lead6$fitted.mean

 E1 <- (lead6$dengue_cases - mu1) / sqrt(mu1)

 N <- nrow(lead6)

 p <- 6 # Max number coef

 dispersion <- sum(E1^2, na.rm=TRUE) / (N - p)

 dispersion <- data.table(date=j, model="ensmeble", disper=dispersion)

 ```

 But this table doesn't seem to be used anywhere in any of the provided

 scripts, so I gather this it is reporting an output statistic.

8. Discussion, page 15: "Relative to a baseline seasonal predictive model (see

 Methods and Materials), the superensemble made, on average, more accurate

 predictions across most provinces, and for most of the year in agreement with

 previous studies [16, 39]. This observation, however, conflicts with the

 results obtained by [30, 42] for whom climate-naïve models had better skill

 than seasonal-climate-informed ones."

 Are there any obvious reasons for this apparent disagreement? Did the

 studies in refs 30 and 42 make use of different data sources, different

 evaluation metrics, etc? And can the authors make any general remarks about

 whether the superensemble forecasts outperformed the climate-naive models in

 refs 30 and 42? I appreciate there are many reasons why a direct comparison

 may not be feasible, and if that's the case then could the authors please

 make this clear in the text so that reader isn't left wondering?

9. Discussion, page 15: "Third, our computations of dengue risk do not take into

 consideration uncertainties due to the potential under- or misreporting of

 dengue cases. Consequently, our model superensemble forecasts may

 underestimate the real number of cases occurring at any given time."

 That's a good point to make. Do the authors have any knowledge about whether

 case ascertainment is likely to have changed substantially over the period

 covered by this study (2002-2019)?

# Minor comments

1. Top of page 3, there is an unexpected character:

 "Large diurnal temperature ranges (¿ 20°C) ..."

2. Top of page 4, there is an unexpected character:

 "In areas where dengue incidence is typically low (e.g., ¡ 10 cases per

 month) ..."

3. Figure S7: consider amending the legend to read "No Economic Value" and

 "Economic Value", so that it is explicit what "value" is being considered.

Reviewer #4: 

This is an interesting paper on an important subject, the development of early warning systems for vector-borne diseases, focusing here on dengue in Vietnam. While the general approach appears reasonable there are a number of clarifications and amendments required. Specific comments are given below.

The model predictions presented by the authors seem useful but they are currently somewhat of a black box. In particular, there are two directions for improving the presentation of the results. The first is based upon standard prequential theory of proper scoring rules and the paper would improve if the authors would decompose the overall prediction error for the different scoring rules into calibration and sharpness, the predictive measures corresponding to bias and variance 

In addition, it would be very useful for this work, as well as future developments, to see the individual contribution to the predictive ability of the different components. For example, are the climate forecasts adding substantially to the predictive ability or the earth observation data correspond to the best part of the ensemble's predictive power?

Also, how accurate climate forecasts are the predictors of seasonal climate, i.e. how good are these forecasts on their own right?

This will be illuminating and could also guide the authors' future efforts in improving this work. It will also shed some light upon the paper's drawbacks, since absence of disease control and related data appears rather important, at least a-priori.

In addition, it will lead to a discussion on issues of model mis-specification, since moderate predictive ability of a covariate (like seasonal climate) but reasonable prediction of a function thereof (such as disease incidence) suggests issues of potential model mis-specification.

A vital aspect of this work is the predictive ability of this approach and the authors claim that this is superior to the competition. However, as far as can be seen in this paper, the comparison is not made with a number of state-of-the-art alternatives, but with a baseline model which is a sub-case of the model class used in the paper. 

Is that an accurate description? 

If so, are the model's conclusions a fair representation of the paper's findings?

This should be clarified and amended as appropriate.

An important clarification should be made with respect to terminology. 

Is model superensemble an alternative name for bayesian model averaging (BMA)?

If so this should be clarified.

Incidentally, BMA essentially started in the fifties with the work of Savage, Lindley and Good so appropriate referencing could be added. 

Non-Bayesian (e.g. AIC-based) model weights are also commonly used, including in ecology where the Burnham and Anderson's 2002 book on multimodel inference is highly influential.

In relation to the above comment, please clarify the definition of the ML weights used in equation 5. The DIC weights are often used as a Bayesian equivalent of the AIC-based ones. However, the most commonly used weights are BIC-based, essentially since they agree asymptotically with standard Bayesian theory. However, ML weights are non-standard and the authors will need to specify how are they calculated and give a reference to their theoretical justification.

The model class that this work is based upon seems reasonable. However, some details need to be added, including the specific parameterisation of the negative binomial distribution and the definition of mu (is it the mean of the log data?)

Reviewer #5: The manuscript presents a Bayesian spatio-temporal model for Dengue

counts, which incorporates weather and land-cover variables. The five

subsets of covariates that produced the best one-to-six-months forecasts

during the period 2007-2016 were combined via Bayesian model averaging.

This superensemble outperformed the ensemble members in terms of both CRPS

and point forecasts (RMSE, MAE). It was eventually used to generate

probabilistic forecasts for the 6 months ahead of April 2020.

Cooperation with key stakeholders from from public health institutions

strengthens the paper and supports transferring scientific progress into

practice.

The paper is well written overall, but I have several questions and remarks

about details of the model and how forecasts are generated.

Major comments

--------------

1. An AR1 prior is used for \\delta_{i,m[t]}. Shouldn't the model assume

 a cyclic effect, i.e., have January conditioned on December?

2. Eq (1): What is the motivation behind scaling past counts by the

 population, i.e., having the terms (1-\\rho) log(P) + \\rho log(Y)?

 I think I haven't seen this in related models before. Did explorative

 analyses indicate a better fit when incorporating past incidence

 instead of past counts?

3. Related to 2.: The model fit and forecasts could probably be improved

 by allowing for a region-specific effect of log(Y_{i,t-1}), i.e.,

 region-specific force of infection \\rho_i. Scaling by the population

 would no longer be necessary and the model would more adequately

 capture regional variation in unobserved important determinants of

 disease spread. A recent example of a related model with

 region-specific force of infection is https://doi.org/10.1002/sim.8390 .

4. I was really confused by how k-step-ahead (k>1) forecasts are generated.

 The manuscript intends to provide superensemble forecasts of regional

 dengue incidence up to 6 months ahead *using seasonal climate forecasts*,

 but then a *climate naive* model is used to generate forecasts for t+2

 to t+6? Why make the (impressive!) effort of collecting climate data and

 forecasts if they are not used to inform probabilistic forecasts?

 Furthermore, how is the forecast distribution generated? Are these

 simple negative binomial forecasts conditioning on the *point*

 forecast for y_{t+1}, ignoring the uncertainty in that forecast? I'm not sure

 if this counts as "properly reflect forecast uncertainty" (discussion).

 Maybe simulation-based path forecasts should be used instead?

5. Eq (3): I think it may be useful to note that the baseline model isn't

 such a simple model after all. The province-specific and time-varying

 random effects will compensate for much of the missing X variables. I

 think a more important "feature" of the baseline model is the missing

 autoregressive term. This omission should also be kept in mind when

 discussing performance relative to this model around p. 15, l. 504. I

 would expect a non-dynamic model to perform particularly well in

 regions with only sporadic dengue cases. Please also report

 the performance of the baseline model in Table 1.

6. One model should really be added to prove the usefulness of the

 superensemble for forecasting: Does the superensemble outperform a model

 which contains all of the covariates of models 1 to 5 combined? If so I'm

 willing to believe that "we demonstrate that using a model superensemble

 results in better forecasts than using individual models" (discussion).

7. Can we conclude from Figure 1 that it would make sense to include a

 non-dynamic baseline model in the ensemble? As Reich et al (2019) put

 it: "building ensemble models that capitalize on the strengths of a

 diverse set of individual component models will be critical to

 improving accuracy and consistency of models in all infectious disease

 forecasting settings." (https://doi.org/10.1073/pnas.1812594116)

8. Table 1: Please report which lags have been selected for the climate

 predictors in the different models. I think this could be an

 interesting result by itself.

9. The ROC analysis was primarily used to compare AUC values across different

 forecast horizons (Figure 5), which is not very interesting. It seems more

 relevant to compare how well the superensemble can forecast the four

 different outbreak indicators (there does not seem to be much difference but

 it is difficult to compare across plots). The text also says that an optimal

 cut-off value for the probability is defined to maximize sensitivity and

 specificity (which is no clear criterion since you cannot maximize both), but

 I cannot find the resulting cut-offs.

Minor comments

--------------

1. Strong underreporting:

 * Intro (first two paragraphs): The question arises why there is such

 strong underreporting. Does dengue seldom require treatment? The intro

 should probably mention common symptoms of dengue infection.

 * At the end, the discussion briefly mentions that the model only

 considers *reported* disease incidence. Thus, routine forecasting

 relies on stable reporting rates. If there were efforts to reduce

 underreporting at some point, endemic channels or quantiles based on

 historical incidence wouldn't make sense any more. Are there any such

 efforts in Vietnam? How could forecasts integrate such a (gradual)

 change?

2. p. 3, l. 82: "deterministic models" are mixed up with "point forecasts"

 here. ARIMA or Poisson regression used in the referenced papers are

 *not* deterministic. The point is that you would like to quantify

 the uncertainty of your forecasts in addition to your best (point)

 estimate of future disease spread.

3. p. 4, l. 102: This sounds like CUSUM methods should be considered,

 see e.g., https://doi.org/10.1016/j.prevetmed.2009.05.017 .

4. Materials and methods should also say where the shapefiles originate.

5. Ref 59 for the Worldpop project contains a wrong URL.

6. Why are two different population datasets [refs 59 and 60] used for P

 in the model and to weight monthly averages of climate variables,

 respectively?

7. p. 5, l. 161: land-cover variables are per province _and per year_

 (please clarify this). I thought they were time-constant until I read

 they vary annually later in the text.

8. p. 5, l. 175: There seem to be two data sources here; where did the

 data eventually come from? Furthermore, the training period ranged from

 2002 to 2006, the testing period from 2007 to 2016, and the forecasting

 period from April 2020 to September 2020. How does hindcast data for

 only 2012-2016 fit in here?

9. p. 6, l. 182: I'd prefer to see "month t = 1, ..., T" instead of the

 more general "time t" to clarify the temporal resolution of the model.

10. On Eq (1): Please specify the distribution assumed for the temporal

 random effects, probably Gaussian.

11. p. 6, l. 188: This is a log-linear model with log-transformed past

 counts, so \\rho is not an autoregressive *coefficient* in the usual

 sense: there is a multiplicative effect (Y_{i,t-1}/P_{i,t-1})^\\rho on

 the mean. Maybe the more general term "autoregressive parameter"

 should be used for \\rho.

12. p. 6, l. 215: I don't understand why annual variation is a reason for

 the land-cover variables to be excluded from the selection procedure.

13. p. 6, l. 225: n is undefined.

14. p. 7, l. 230: "the best two performing models for each verification

 metric" doesn't seem correct. The results show only 5 models, so it

 seems only the top model for each verification metric has been

 selected. But then in Table 1, model 2 seems to be best for both RMSE

 and MAE, so how obtain five component models? Please clarify the

 selection procedure.

15. Eq (4) and (5): It seems there is some confusion with indices i and k.

 For example, M_k in (4) should be M_i. Furthermore, why is w_i a

 vector and how does it sum to 1 given (5)? The "ML or DIC" part isn't

 quiet clear.

16. Eq (6): CRPS_f and CRPS_b are undefined.

17. p. 9, l. 314-330: This description is repetitive and should be

 dropped. The results section should concentrate on results, not

 methods.

18. Figure 7: Shouldn't this be based on the 95% instead of the 75% quantile

 since the former had better scores?

[LINK]

---

## [Decision Letter · Decision Letter 2]

20 Nov 2020

Dear Dr. Colón-González,

Thank you very much for submitting your revised manuscript "Probabilistic seasonal dengue forecasting in Vietnam: A modelling study using superensembles" (PMEDICINE-D-20-02989R2) for consideration at PLOS Medicine. 

Your paper was once again evaluated by a senior editor and discussed among all the editors here. It was also sent to three of the previous independent reviewers, including a statistical reviewer. The reviews are appended at the bottom of this email and any accompanying reviewer attachments can be seen via the link below:

[LINK]

In light of these reviews, we would like to once more consider a revised version that addresses the reviewers' and editors' comments. Obviously we cannot make any decision about publication until we have seen the revised manuscript and your response, and we may seek re-review by one or more of the reviewers. 

We expect to receive your revised manuscript by Dec 11 2020 11:59PM. Please email us (plosmedicine@plos.org) if you have any questions or concerns.

We look forward to receiving your revised manuscript. 

Sincerely,

Artur Arikainen, 

Associate Editor 

PLOS Medicine

plosmedicine.org

1. Please address the reviewer’s additional comments below on the description of your methodology.

2. Please include line numbers in the margin throughout your text, including the abstract and author summary.

3. Abstract: Please include some summary quantitative data on the fit of your model to prior data and on your future predictions (with 95% CIs and p values, as appropriate).

---

Comments from the reviewers:

Reviewer #2: Thank you for addressing my previous comments. It's a nice article, well done.

Reviewer #4: The authors have revised their paper and this now represents a substantially improved manuscript, acceptable for publication

Reviewer #5: My main concern remains how k-step ahead forecasts are generated. Since forecasting is the main purpose of this manuscript, I think the corresponding description should be improved.

I admit that I didn't have the time to look at the revised version in every detail. Nevertheless, I stumbled over a few issues in the authors' reply, which I would like to bring up for consideration.

1. Eq 3+4: The math notation is too sloppy for a Bayesian inference approach. I've preferred the previous simplistic textual description of assuming a NegBin for Y_it. Now a random variable $Y$ (without $_{i,t}$) is introduced, whose meaning is undefined, and appending y >= 0 doesn't make any sense. This Bayesian model should be either formulated with mathematical rigour, or with just the core part, i.e., Eq. 4, i.e., the conditional mean of $Y_{i,t}$.

2. Eq. 3 would imply that this paper assumes a marginal NegBin for $Y_{i,t}$, but my previous reading suggests that a conditional NegBin was assumed, in particular also conditioning on the past observations just like for other covariates. Please clarify.

3. Eq. 4 now says that $\\rho$ works on past counts not past incidence. However, the text below still contains Y/P. Please check.

4. The reply confirms that k-step-ahead (k>1) forecasts as implemented here do not carry forward the uncertainty of one-step-ahead forecasts. My interpretation is that the authors eventually assume a NegBin distribution for their forecast $Y_{t+2} | Y_t, ...$, by *imputing* a point prediction for $Y_{t+1}$ to actually compute $Y_{t+2} | Y_{t+1}, ...$. AFAIK, $Y_{t+2} | Y_t$ follows a *mixture* of NegBin distributions (which is not necessarily NegBin), see, e.g., https://doi.org/10.1016/j.ijforecast.2020.07.002 (Section 3.4) for a correct approach. I understand that it is too late to fix this, but it would still be possible to include a cautious comment that all forecasts (including for k > 1) assume a conditional NegBin distribution, imputing one-step-ahead point predictions for lagged counts. I don't know if there is a name for such "naive" forecasts.

5. Table 1 should use two digits in the Bias column.

6. A "ROC curve" analysis would be used to choose a diagnostic threshold and display its effect on TPR and FPR and to display the overall diagnostic accuracy as the area under the ROC curve. I'm thus puzzled by the use of the term "ROC curve" analysis in the manuscript.

---

[LINK]

---

## [Decision Letter · Decision Letter 3]

12 Jan 2021

Dear Dr. Colón-González,

Thank you very much for re-submitting your manuscript "Probabilistic seasonal dengue forecasting in Vietnam: A modelling study using superensembles" (PMEDICINE-D-20-02989R3) for review by PLOS Medicine.

I have discussed the paper with my colleagues and the academic editor and it was also seen again by one of the reviewers. I am pleased to say that provided the remaining editorial and production issues are dealt with we are planning to accept the paper for publication in the journal.

[LINK]

We look forward to receiving the revised manuscript by Jan 19 2021 11:59PM.   

Sincerely,

Artur Arikainen, 

Associate Editor 

PLOS Medicine

plosmedicine.org

Requests from Editors:

1. Data Availability Statement (DAS): Note that a study author cannot be the contact person for a dataset. Please provide alternative details for your dengue data.

2. Abstract: Please ensure that all quantitative data are also presented in the main Results section, and that the values match.

3. Throughout, please perhaps replace “skill” with a more appropriate term (eg. ‘strength’), when describing the performance of the forecast system.

4. In the introduction, please mention dengue vaccination - although a complex and not very satisfactory situation, this could be discussed.

5. Please mention in your Methods that your study did not require separate ethical approval.

6. Line 749: "explicitly incorporate".

7. Please provide a URL or DOI for reference 6.

8. RE reference 91 listed as ”in press”, papers cannot be listed in the reference list until they have been accepted for publication or are otherwise publicly accessible (for example, in a preprint archive). The information may be cited in the text as a personal communication with the author if the author provides written permission to be named. Alternatively, please provide a different appropriate reference. 

Comments from Reviewers:

Reviewer #5: Thank you for addressing my previous comments, especially for clarifying k-step-ahead as imputed one-step-ahead forecasts and including cautious comments about that.

In my opinion, this manuscript is now ready for publication. Congratulations!

[LINK]

---

## [Editor Report · Decision Letter 4]

22 Jan 2021

Dear Dr Colón-González, 

On behalf of my colleagues and the guest Academic Editor, Alex R. Cook, I am pleased to inform you that we have agreed to publish your manuscript "Probabilistic seasonal dengue forecasting in Vietnam: A modelling study using superensembles" (PMEDICINE-D-20-02989R4) in PLOS Medicine.

PRESS

Sincerely, 

Artur A. Arikainen 

Associate Editor 

PLOS Medicine